# Molecular insights into intrinsic transducer-coupling bias in the CXCR4-CXCR7 system

Parishmita Sarma[1], Carlo Marion C. Carino[2], Deeksha Seetharama[1], Shubhi Pandey[1], Hemlata Dwivedi-Agnihotri[1], Xue Rui[3], Yubo Cao [4], Kouki Kawakami [2], Poonam Kumari[5], Yu-Chih Chen[6], Kathryn E. Luker [7], Prem N. Yadav[5], Gary D. Luker[7,8], Stéphane A. Laporte [4,9], Xin Chen[3], Asuka Inoue [2] & Arun K. Shukla [1] ✉

Chemokine receptors constitute an important subfamily of G protein-coupled receptors (GPCRs), and they are critically involved in a broad range of immune response mechanisms. Ligand promiscuity among these receptors makes them an interesting target to explore multiple aspects of biased agonism. Here, we comprehensively characterize two chemokine receptors namely, CXCR4 and CXCR7, in terms of their transducer-coupling and downstream signaling upon their stimulation by a common chemokine agonist, CXCL12, and a small molecule agonist, VUF11207. We observe that CXCR7 lacks G-protein-coupling while maintaining robust βarr recruitment with a major contribution of GRK5/6. On the other hand, CXCR4 displays robust G-protein activation as expected but exhibits significantly reduced βarr-coupling compared to CXCR7. These two receptors induce distinct βarr conformations even when activated by the same agonist, and CXCR7, unlike CXCR4, fails to activate ERK1/2 MAP kinase. We also identify a key contribution of a single phosphorylation site in CXCR7 for βarr recruitment and endosomal localization. Our study provides molecular insights into intrinsic-bias encoded in the CXCR4-CXCR7 system with broad implications for drug discovery.

Chemokines are small secreted proteins that typically exert their actions via chemokine receptors belonging to the large superfamily of G protein-coupled receptors (GPCRs), also known as seven transmembrane receptors (7TMRs)[1,2]. Chemokines and chemokine receptors contribute to a diverse array of physiological processes, especially in various aspects of immune response activation and regulation[2,3]. A peculiar aspect in the chemokine-chemokine receptor system is ligand promiscuity where not only a single chemokine can bind to, and activate multiple chemokine receptors, but a given receptor can also be activated by several different chemokines[4,5]. Chemokine receptors typically couple to, and signal through, heterotrimeric G-proteins and β-arrestins (βarrs), as expected for prototypical GPCRs[6]. Interestingly however, there are several examples of chemokine receptors that exhibit a significant deviation from this paradigm, especially with

[1]Department of Biological Sciences and Bioengineering, Indian Institute of Technology, Kanpur 208016, India. [2]Graduate School of Pharmaceutical Sciences, Tohoku University, Sendai, Miyagi 980-8578, Japan. [3]Department of Medicinal Chemistry, School of Pharmaceutical Engineering and Life Science, Changzhou University, Changzhou, Jiangsu 213164, China. [4]Department of Pharmacology and Therapeutics, McGill University, Montréal, QC H3G 1Y6, Canada. [5]Neuroscience and Ageing Biology Division, CSIR-Central Drug Research Institute Sector 10, Sitapur Road, Lucknow 226031 Uttar Pradesh, India. [6]Department of Computational and Systems Biology, Department of Bioengineering, University of Pittsburgh, Pittsburgh, PA, USA. [7]Center for Molecular Imaging, Department of Radiology, University of Michigan, Ann Arbor, MI, USA. [8]Department of Biomedical Engineering, Department of Microbiology and Immunology, University of Michigan, Ann Arbor, MI, USA. [9]Department of Medicine, McGill University Health Center, McGill University, Montréal, QC H4A 3J1, Canada. ✉e-mail: arshukla@iitk.ac.in

respect to their transducer-coupling patterns and downstream signaling responses[7–11].

The CXC chemokine receptor subtype 4 (CXCR4) and subtype 7 (CXCR7; also known as Atypical Chemokine Receptor 3, ACKR3) constitute an interesting pair as they both recognize a common natural chemokine agonist, referred to as CXCL12, also known as the stromal cell-derived factor 1 (SDF1)[12] (Fig. 1a) These two receptors are involved

in various aspects of cancer onset and progression, cardiac disorders and autoimmune diseases[13]. Interestingly, CXCR7, but not CXCR4, also recognizes another chemokine referred to as CXCL11[14]. CXCR4 is widely considered a prototypical GPCR with coupling to Gαi subfamily of G-proteins as measured in terms of inhibition of cAMP, and it also recruits βarrs upon agonist-stimulation[15]. On the other hand, stimulation of CXCR7 by CXCL12 or CXCL11 fails to elicit any measurable Gαi

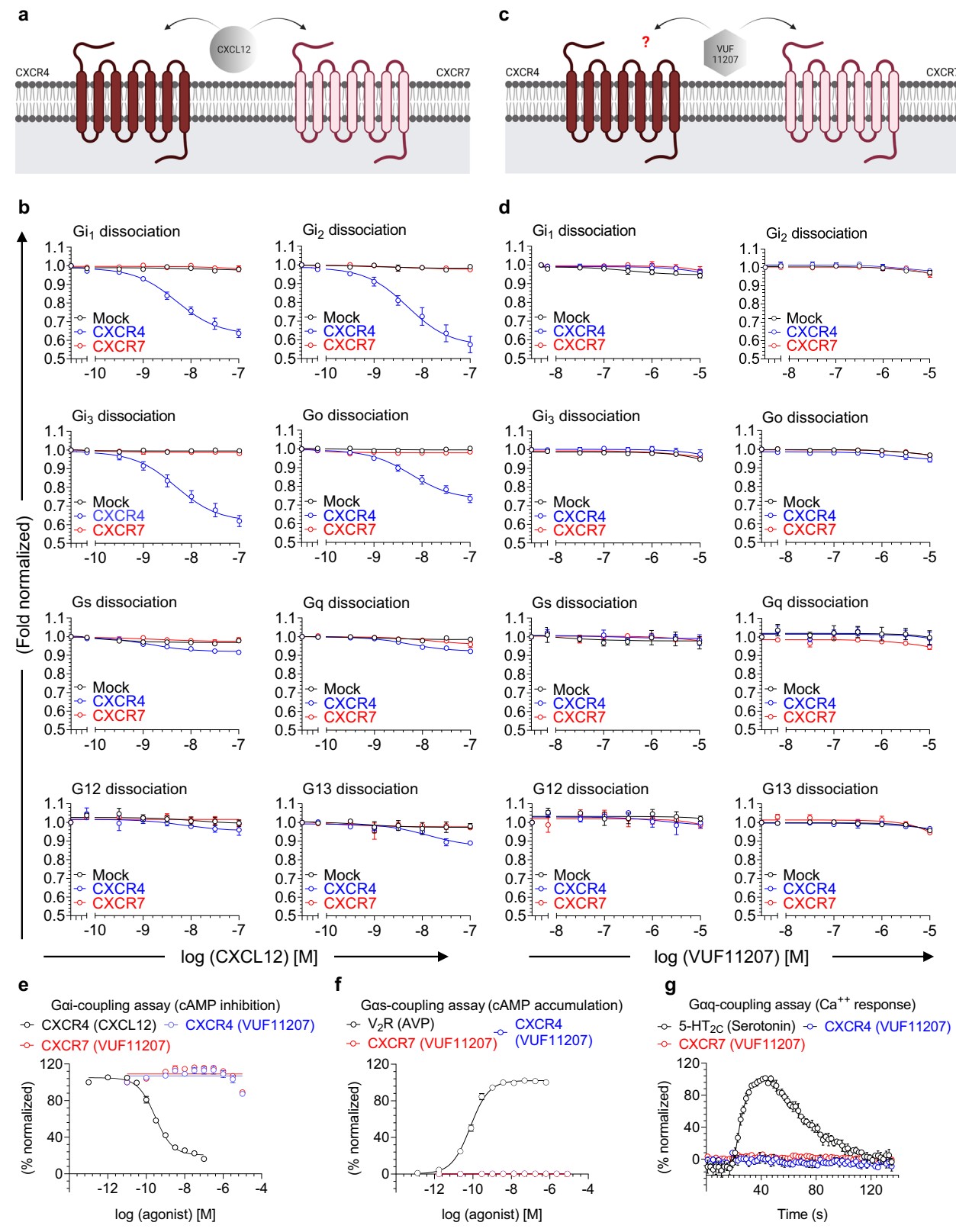

**Fig. 1 | Lack of G-protein activation upon stimulation of CXCR7. a** CXCL12, a CXC type chemokine, is a common agonist for both, CXCR4 and CXCR7 (created with BioRender.com). **b** NanoBiT-based assay for CXCL12-induced dissociation of heterotrimeric G-proteins for CXCR4 and CXCR7 (mean ± SEM; $n = 4$-5 independent experiments; i.e., for Gi$_1$ dissociation: Mock, $n = 4$; CXCR4, $n = 5$; CXCR7, $n = 4$; for Gi$_2$ and Gi$_3$ dissociation: $n = 5$; for Go dissociation: Mock, $n = 4$; CXCR4, $n = 5$; CXCR7, $n = 5$; for Gs, Gq, G12, and G13 dissociation: $n = 4$; normalized with luminescence signal under unstimulated condition taken as 1). Mock represents empty vector transfected cells as a negative control. **c** VUF11207 is a small molecule agonist for CXCR7 but its efficacy for CXCR4, if any, is not known (created with BioRender.com). **d** NanoBiT-based assay for VUF11207-induced dissociation of heterotrimeric G-proteins for CXCR4 and CXCR7 (mean ± SEM; $n = 3$ independent experiments; normalized with luminescence signal under unstimulated condition taken as 1). **e** Agonist-induced decrease in forskolin-induced cAMP level measured using the GloSensor assay for the indicated receptor-ligand combinations as a readout of Gαi-activation (mean ± SEM; $n = 4$; normalized with the signal at minimal ligand dose for CXCL12-CXCR4 combination as 100%). **f** Agonist-induced increase in cAMP level measured using the GloSensor assay for the indicated receptor-ligand combinations as a readout of Gαs-activation (mean ± SEM; $n = 3$; normalized with maximal signal for V2R as 100%). V2R (vasopressin receptor subtype 2) is used as a positive control. **g** Agonist-induced increase in Ca$^{++}$ level measured using the GCaMP sensor for the indicated receptor-ligand combinations as a readout of Gαq-activation (mean ± SEM; $n = 4$; normalized with maximal signal for serotonin as 100%). 5-HT2C receptor is used as a positive control. Source data are provided as a source data file.

activation, although there are reports that suggest its ability to recruit βarrs[16,17]. Interestingly, a small molecule ligand known as VUF11207 has also been reported to promote CXCR7-βarr interaction[18], although its interaction with other CXCRs and complete transducer-coupling profile has not been evaluated thus far. Therefore, the CXCR4-CXCR7 pair represents an intriguing system to probe the molecular and structural details of intrinsic transducer-coupling bias.

Here, we present a comprehensive investigation of agonist-induced G-protein-coupling, contribution of GRKs in βarr recruitment and conformational signatures, and ERK1/2 MAP kinase activation downstream of CXCR4 and CXCR7 using the shared natural agonist CXCL12 and a small molecule compound, VUF11207. Our study provides the molecular details of the intrinsic bias encoded in the CXCR4-CXCR7 system and their functional divergence. These findings not only offer important insights to better understand biased agonism at 7TMRs but also present an experimental framework that may guide analogous exploration of other chemokine receptors.

## Results

### Agonist-induced G-protein-activation and second messenger response

CXCR7, unlike CXCR4, is considered to lack G-protein-coupling in response to CXCL12 stimulation, although the experimental evidence is limited primarily to the lack of canonical Gαi-activation[17]. Therefore, we first set out to comprehensively probe CXCL12-induced G-protein activation using a NanoBiT-based heterotrimer dissociation assay[19] for CXCR4 and CXCR7. In this assay, agonist-stimulated G-protein activation is measured as a decrease in luminescence signal arising from dissociation of the NanoBiT-engineered heterotrimer consisting of the Gα-LgBiT, SmBiT-Gγ2 and untagged Gβ1 subunits. We observed that CXCR4 robustly activates G-proteins of the Gαi subfamily but CXCR7 remains silent in this assay not only for Gαi but other subtypes as well (Fig. 1b). As mentioned earlier, a small molecule ligand has also been described for CXCR7 although its characterization remains limited to binding studies and βarr recruitment in a BRET assay (Fig. 1c). We therefore decided to also test VUF11207 in the NanoBiT-based heterotrimer dissociation assay to probe its ability to activate G-proteins, if any. We observed that like CXCL12, VUF11207 also fails to elicit any measurable G-protein activation from CXCR7 (Fig. 1d). Moreover, VUF11207 also did not promote any G-protein activation for CXCR4, suggesting its selectivity for CXCR7 (Fig. 1d). In these assays, surface expression of both the receptors was optimized to be at comparable levels as measured using a flow-cytometry based assay (Supplementary Fig. 1a).

We also tested VUF11207 in second messenger assays based on cAMP response and calcium release, and did not observe a measurable response for CXCR7, which further confirms the inability of CXCR7 to activate G-proteins (Fig. 1e–g). A previous study has reported that stimulation of HEK-293 cells with CXCL12 results in measurable calcium mobilization due to endogenous CXCR4[20]. Thus, the absence of calcium response upon VUF11207 stimulation further indicates the lack

of its binding and/or agonism at CXCR4, which is further corroborated by the absence of any measurable response even upon CXCR4 over-expression (Fig. 1e–g). We also measured surface expression of CXCR4 and CXCR7 in these assays, which was always higher than mock-transfected cells although in some experiments, CXCR7 expression was only marginally above the mock-transfection (Supplementary Fig. 1b, c). Therefore, we repeated these experiments with higher expression levels of CXCR7 by increasing the amount of transfected DNA but still did not observe any response for CXCR7 (Supplementary Fig. 1d, e). Taken together, these data establish the inherent inability of G-protein activation by CXCR7 upon stimulation by CXCL12 or VUF11207.

### Agonist-induced βarr recruitment

Although previous studies have shown βarr coupling to CXCR4[15] and CXCR7[16], a comprehensive side-by-side analysis of recruitment of both βarr isoforms i.e. βarr1 and 2, to both receptors has not been described thus far. Therefore, we used two different assays to measure recruitment of βarr1 and 2 to both these receptors in response to CXCL12 and VUF11207. First, we measured CXCL12-induced βarr2 recruitment in PRESTO-Tango assay[21], where the receptors are engineered to contain the carboxyl-terminus of vasopressin subtype 2 receptor (i.e. CXCR4-V$_2$ and CXCR7-V$_2$). While we observed a robust response for CXCR7 in a ligand dose-dependent manner, the basal luminescence signal in CXCR4-expressing cells was high and it increased only slightly in response to CXCL12 (Fig. 2a and Supplementary Fig. 2a). Considering that PRESTO-Tango constructs use a chimeric receptor, we also generated new Tango assay constructs for CXCR4 and CXCR7 without the V$_2$R carboxyl-terminus fusion, and measured βarr2 recruitment in response to CXCL12. We observed a robust agonist-induced response for CXCR7, however, CXCR4 displayed a significantly lower E$_{max}$ for βarr2 recruitment compared to CXCR7 (Fig. 2b and Supplementary Fig. 2b). On the other hand, VUF11207 effectively promoted βarr2 recruitment for CXCR7 but not for CXCR4 further suggesting its specificity at CXCR7 (Fig. 2c, d). We also noted that CXCR7 expression in PRESTO-Tango assay was only marginally higher than mock-transfection (Supplementary Fig. 2a), and therefore, we repeated these experiments in cells transfected with higher amount of CXCR7-V$_2$ plasmid. While the surface expression of CXCR7 improved only moderately over mock-transfection (Supplementary Fig. 2c), βarr2 recruitment nearly doubled in these experiments (Supplementary Fig. 2d) further corroborating the ability of CXCR7 to efficiently recruit βarr2.

We found the weaker βarr2 recruitment response for CXCR4 compared to CXCR7 in the Tango assay intriguing, and therefore, we generated NanoBiT assay constructs for CXCR4 and CXCR7, which enable us to measure βarr response in real time, to probe this further. Here, we generated and tested several combinations of CXCR4 and βarr1/2 NanoBiT constructs (i.e. CXCR4-SmBiT+LgBiT-βarr1/2 vs. βarr1/2-LgBiT, and CXCR4-LgBiT+SmBiT-βarr1/2 vs. βarr1/2-SmBiT), and we observed that CXCR4-SmBiT+LgBiT-βarr1/2 combination yielded

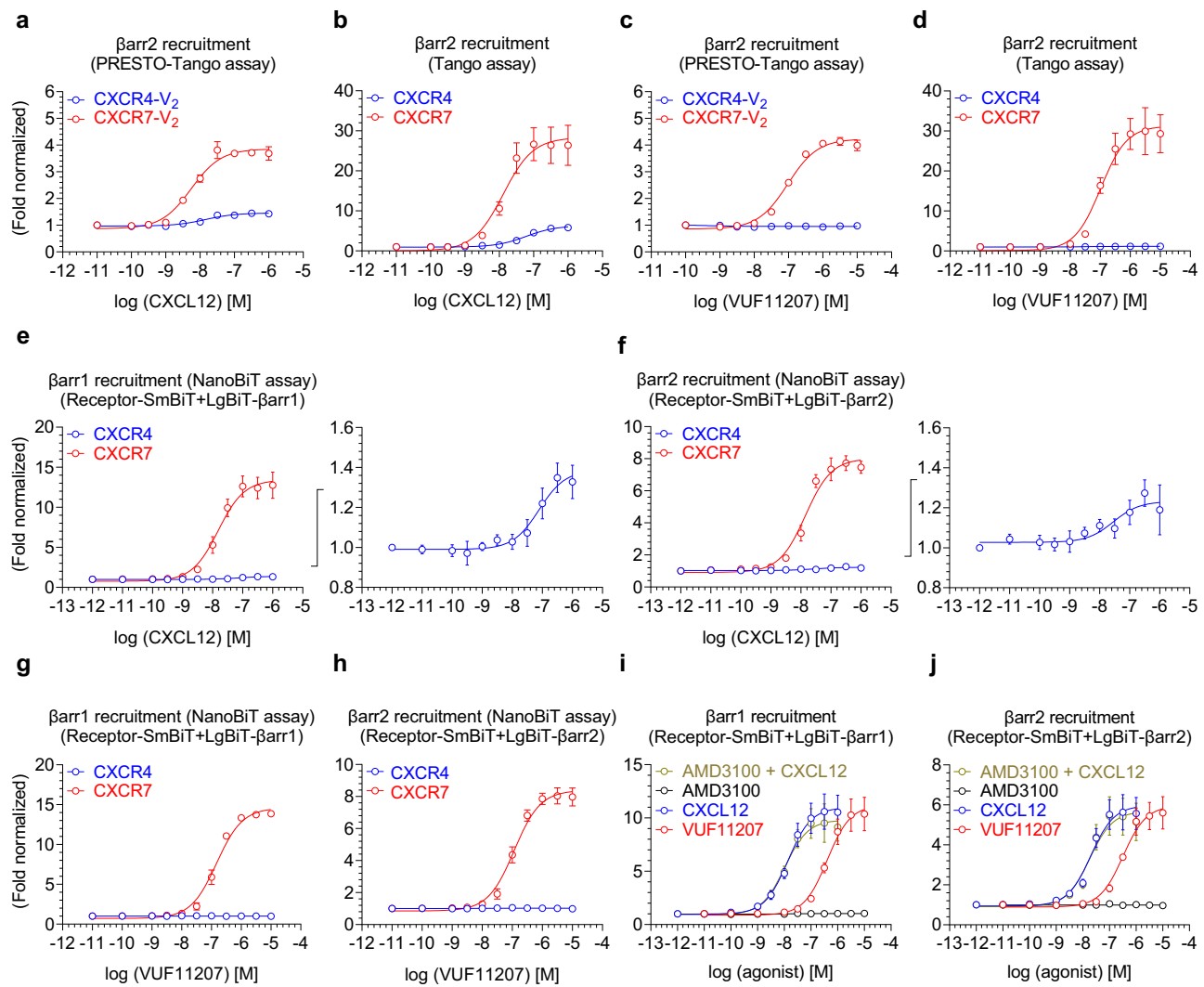

**Fig. 2 | β-arrestin recruitment to CXCR7. a**, **b** CXCL12-induced βarr2 recruitment to CXCR4 and CXCR7 in PRESTO-Tango and Tango assays, respectively (mean ± SEM; $n = 3$ independent experiments; normalized with the luminescence signal at minimal ligand dose treated as 1). The PRESTO-Tango assay uses a chimeric receptor construct with the carboxyl-terminus of $V_2R$ while the Tango assay uses native receptors. **c**, **d** VUF11207-induced βarr2 recruitment to CXCR4 and CXCR7 in PRESTO-Tango and Tango assays, respectively (mean ± SEM; $n = 3$ independent experiments; normalized with the luminescence signal at minimal ligand dose treated as 1). **e**, **f** CXCL12-induced βarr1/2 recruitment to CXCR4 and CXCR7 in NanoBiT assay (mean ± SEM; $n = 4$ independent experiments; normalized with

luminescence signal at minimal ligand dose treated as 1). Response for CXCR4 is also shown separately in the right panels. **g**, **h** VUF11207-induced βarr1/2 recruitment to CXCR4 and CXCR7 in NanoBiT assay (mean ± SEM; $n = 4$ independent experiments; normalized with luminescence signal at minimal ligand dose treated as 1). **i**, **j** A side-by-side comparison of CXCL12- vs. VUF11207-induced βarr1 and 2 recruitment to CXCR7, respectively (mean ± SEM; $n = 4$ independent experiments; normalized with luminescence signal at minimal ligand dose treated as 1). A CXCR4-specific antagonist AMD3100 is used either alone, or as pre-treatment to CXCL12, as a negative control and to rule out the possibility of any contribution from endogenous CXCR4. Source data are provided as a source data file.

maximal response although it was still only two-fold over basal signal (Supplementary Fig. 3a–f). Subsequently, we used CXCR4/CXCR7-SmBiT+LgBiT-βarr1/2 combination to measure agonist-induced βarr recruitment for the two receptors. Similar to Tango assay data, we observed a stronger recruitment in terms of $E_{max}$ of βarr1 and 2 for CXCR7 compared to CXCR4 upon stimulation with CXCL12 (Fig. 2e, f and Supplementary Fig. 2e, f) while VUF11207 selectively promoted βarr1 and 2 recruitment to CXCR7 but not to CXCR4 (Fig. 2g, h). These data not only suggest the selectivity of VUF11207 for CXCR7 but more importantly, also underscore a relatively lower propensity of βarr-coupling to CXCR4 compared to CXCR7. We acknowledge that we cannot unequivocally rule out the possibility that βarr recruitment to CXCR4 is comparable to that of CXCR7, however, differential conformational restraints in βarrs lead to poor efficiency of protease cleavage and fragment complementation in the Tango and NanoBiT assays, respectively, resulting in weaker response. Still however, the

relatively weaker signal for βarr recruitment upon stimulation of CXCR4 is intriguing, especially considering that CXCR4 harbors several potential phosphorylation sites in the carboxyl-terminus, and they have been implicated in agonist-induced βarr recruitment[20,22]. Therefore, additional studies are warranted to further probe this interesting observation with respect to receptor phosphorylation and distinct modes of βarr engagement[23–27].

A closer analysis of βarr recruitment to CXCR7 in the NanoBiT assay suggested a difference in the $EC_{50}$ between CXCL12 and VUF11207. Therefore, we compared these two ligands side-by-side and confirmed that CXCL12 is more potent for βarr recruitment over VUF11207, although their $E_{max}$ values are comparable (Fig. 2i, j). In order to rule out the contribution of endogenous CXCR4 in relatively higher potency of CXCL12 compared to VUF11207, we pre-treated the cells with AMD3100, a CXCR4 antagonist, followed by CXCL12 stimulation. However, we did not observe any change in

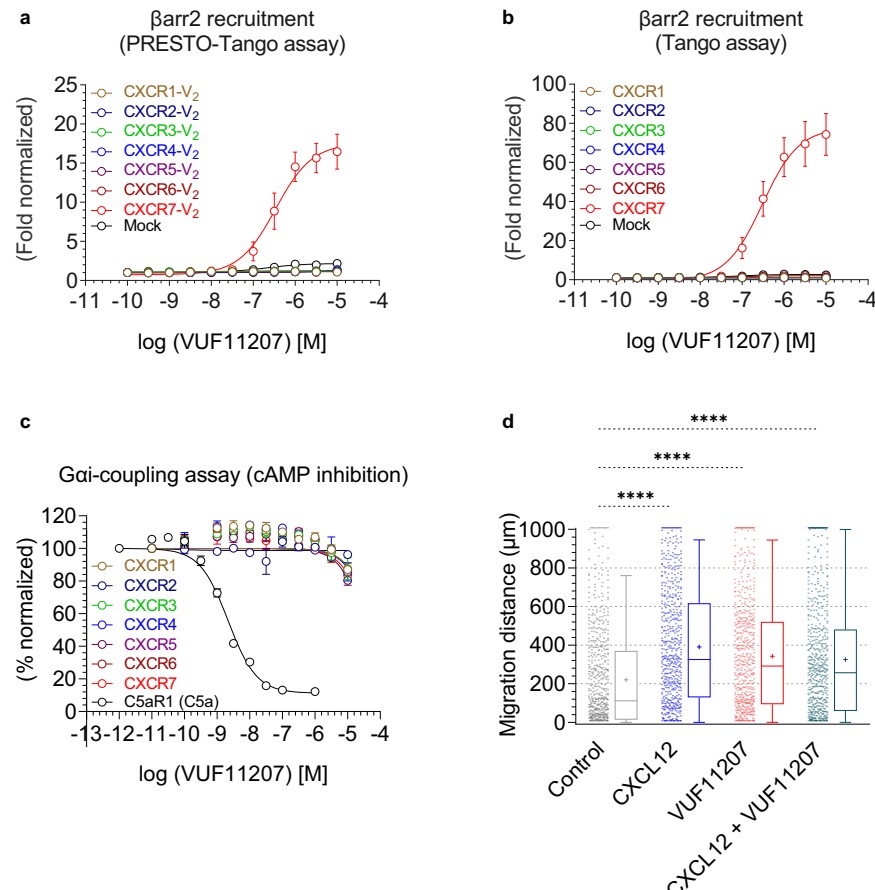

**Fig. 3 | Receptor sub-type selectivity of VUF11207 and its ability to promote cell migration. a**, **b** VUF11207-induced βarr2 recruitment for all the CXC chemokine receptors (CXCR1-7) in the PRESTO-Tango and Tango assay, respectively (mean ± SEM); *n* = 5 independent experiments; normalized with the luminescence signal at minimal ligand dose treated as 1. **c** VUF11207-induced Gαi-coupling for all the CXC chemokine receptors (CXCR1-7) in the GloSensor assay (mean ± SEM; *n* = 3 independent experiments; normalized with luminescence signal at minimal ligand dose treated as 100%). **d** Migration of MDA-MB-231 stably expressing CXCR7 in response to indicated ligands. The experiment was carried out using a 2D microfluidic device and each point on the graph represents an individual cell (*n* = 600 cells examined

over 2 independent experiments, One-way ANOVA, Šídák's multiple comparisons test). The exact *p*-values are as follows: Control vs. CXCL12 (*p* < 0.0001), Control vs. VUF11207 (*p* < 0.0001), Control vs. CXCL12 + VUF11207 (*p* < 0.0001). The lower and upper whiskers represent the 5th percentile and the 95th percentile respectively and the bounds of box correspond to 25th and 75th percentile. The cross inside the box indicates the mean (Control: 220.798, CXCL12: 390.201, VUF11207: 341.980, CXCL12 + VUF11207: 325.656). The solid line at the 50th percentile indicates the median (Control: 111.150, CXCL12: 325.000, VUF11207: 291.850, CXCL12 + VUF11207: 257.07)5. (****$p$ < 0.0001). Source data are provided as a source data file.

CXCL12-induced βarr1/2 recruitment, and AMD3100 also did not elicit any response by itself as expected (Fig. 2i, j).

### Selectivity profiling of VUF11207 on CXCRs

Inspired by the selectivity of VUF11207 for CXCR7 over CXCR4, we decided to test it on other CXCRs as well. There are seven CXCRs (CXCR1-CXCR7) in the human genome, and we measured VUF11207 response for all of them in parallel using the GloSensor assay for Gαi-coupling and PRESTO-Tango/Tango assay for βarr2 coupling. We observed that VUF11207 was able to induce βarr2 recruitment only for CXCR7 and no other CXCRs (Fig. 3a, b). Moreover, VUF11207 was completely silent for every CXCR tested in the GloSensor-based cAMP assay as a readout of Gαi-coupling (Fig. 3c). We observed some variations in the relative receptor expression in these assays, although they all expressed at levels that are higher than the mock-transfected cells (Supplementary Fig. 4a–c). Taken together, these findings establish VUF11207 as a CXCR7 selective agonist and therefore, provides a pre-validated tool compound to probe the structure and function of CXCR7 in future studies.

Previous studies have reported that both, CXCR4 and CXCR7 promote migration and invasion of several cancer cell lines upon

activation by CXCL12[13,28]. Considering the selectivity of VUF11207 for CXCR7 over CXCR4, we envisioned that it may be a powerful tool to specifically measure the contribution of CXCR7 in migration and invasion of cancer cells. Therefore, we measured the migration of MDA-MB-231 breast cancer cells that were stably transfected with CXCR7 in response to VUF11207 using a 2D microfluidic device[29]. We observed that stimulation of these cells with VUF11207 resulted in efficient migration as compared to vehicle-treated cells, and that VUF11207 response was comparable to that of CXCL12 (Fig. 3d). Simultaneous addition of CXCL12 and VUF11207 did not result in any synergistic effect on migration of these cells (Fig. 3d). Taken together with the striking sub-type selectivity of VUF11207, these data provide direct evidence for a contribution of CXCR7 in the migration of breast cancer cells that aligns with previous studies[30]. It would be interesting to further probe the mechanistic aspect of this observation, for example, with respect to the interplay of CXCR4 and CXCR7 and the contribution of βarr signaling.

### Contribution of GRKs in βarr recruitment

As receptor phosphorylation is a key determinant of βarr recruitment, we next tested the contribution of different GRKs in

 

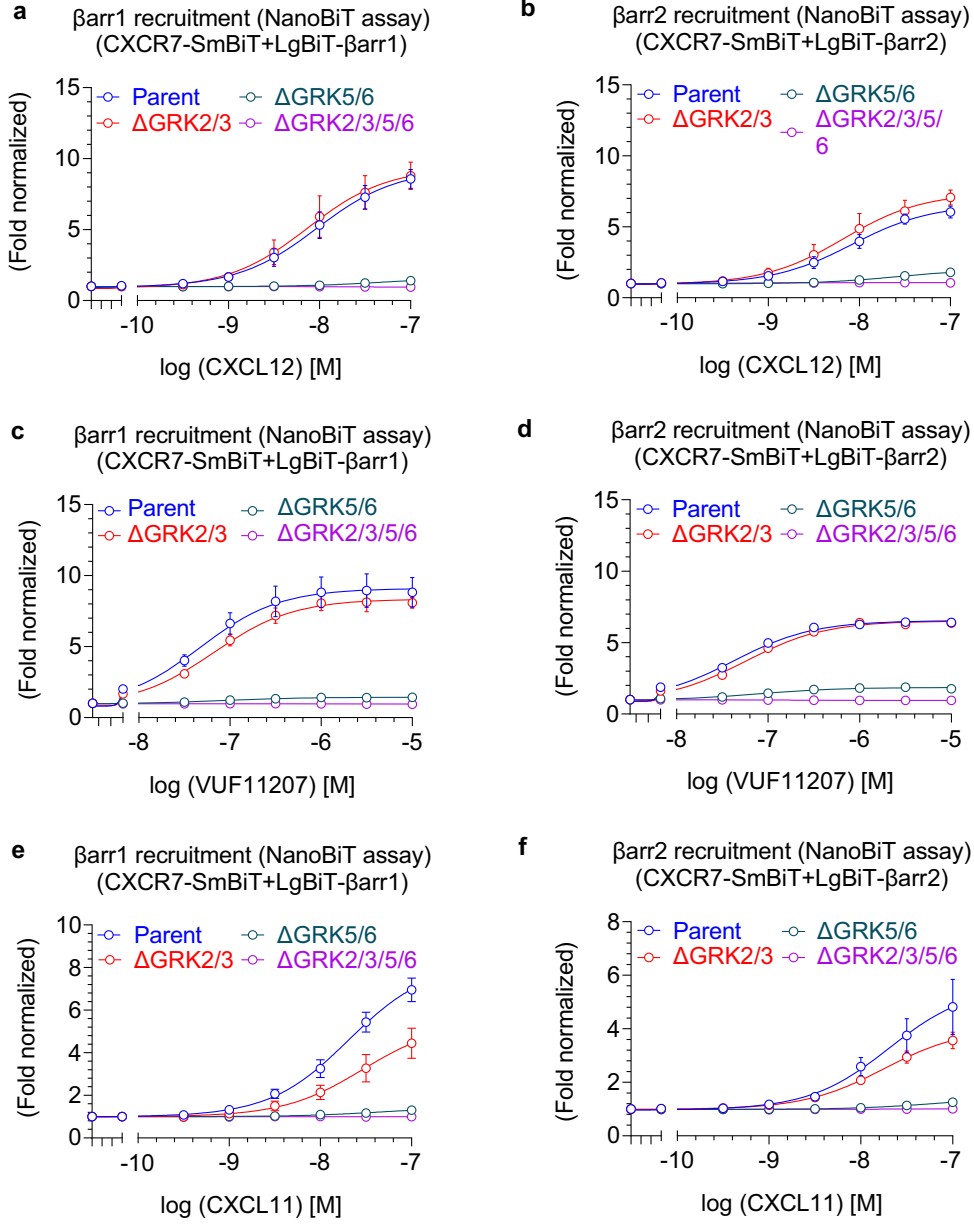

**Fig. 4 | Contribution of GRKs in β-arrestin1/2 recruitment to CXCR7.**
**a**, **b** CXCL12-induced βarr1/2 recruitment to CXCR7 in GRK knock-out cells using the NanoBiT assay (mean ± SEM; $n = 4$–5 independent experiments; i.e., for βarr1 recruitment: Parent, $n = 5$; ΔGRK2/3, $n = 4$; ΔGRK5/6, $n = 4$; ΔGRK2/3/5/6, $n = 5$; for βarr2 recruitment: Parent, $n = 5$; ΔGRK2/3, $n = 4$; ΔGRK5/6, $n = 4$; ΔGRK2/3/5/6, $n = 4$; normalized with luminescence signal under unstimulated condition treated as 1). **c**, **d** VUF11207-induced βarr1/2 recruitment to CXCR7 in GRK knock-out cells using the NanoBiT assay (mean ± SEM; $n = 4$–5 independent experiments; i.e., for βarr1 recruitment: Parent, $n = 5$; ΔGRK2/3, $n = 4$; ΔGRK5/6, $n = 4$; ΔGRK2/3/5/6, $n = 5$; for βarr2 recruitment: Parent, $n = 5$; ΔGRK2/3, $n = 4$; ΔGRK5/6, $n = 4$; ΔGRK2/3/5/6, $n = 4$; normalized with luminescence signal under unstimulated condition treated as 1). **e**, **f** CXCL11-induced βarr1 and 2 recruitment to CXCR7 in GRK knock-out cells using the NanoBiT assay (mean ± SEM; $n = 3$–4 independent experiments; i.e., for βarr1 recruitment: Parent, $n = 3$; ΔGRK2/3, $n = 4$; ΔGRK5/6, $n = 4$; ΔGRK2/3/5/6, $n = 4$; for βarr2 recruitment: Parent, ΔGRK2/3, ΔGRK5/6, and ΔGRK2/3/5/6, $n = 4$; normalized with luminescence signal under unstimulated condition treated as 1). Source data are provided as a source data file.

agonist-induced βarr recruitment to CXCR7 using GRK knock-out cell lines[31]. We observed that knock-out of GRK5/6 nearly ablates CXCR7-βarr1/2 interaction in response to both agonists, CXCL12 and VUF11207 while knock-out of GRK2/3 did not influence βarr recruitment to CXCR7 (Fig. 4a–d). In addition to CXCL12, another chemokine agonist CXCL11 also binds and activates CXCR7[17], and therefore, we also measured the effect of GRK knock-out on CXCL11-induced βarr1/2 recruitment for CXCR7. As presented in Fig. 4e, f, CXCL11-induced βarr1/2 recruitment was also sensitive primarily to GRK5/6 knock-out although GRK2/3 knock-out also appears to have some effect on βarr recruitment, which is more pronounced for

βarr1 than βarr2. These observations converge with the lack of G-protein-coupling to CXCR7 because GRK2/3 translocation to the plasma membrane and activation has previously been shown to require Gβγ release[32]. We also assessed agonist-induced CXCR7-βarr interaction in presence of pertussis toxin (PTX), however, the pattern of recruitment in response to either of the agonists did not change significantly (Supplementary Fig. 5a–f). This observation rules out activation-independent contribution of Gαi on βarr recruitment, and therefore, establishes CXCR7 as a model βarr-coupled receptor system to further investigate the functional contribution of βarrs.

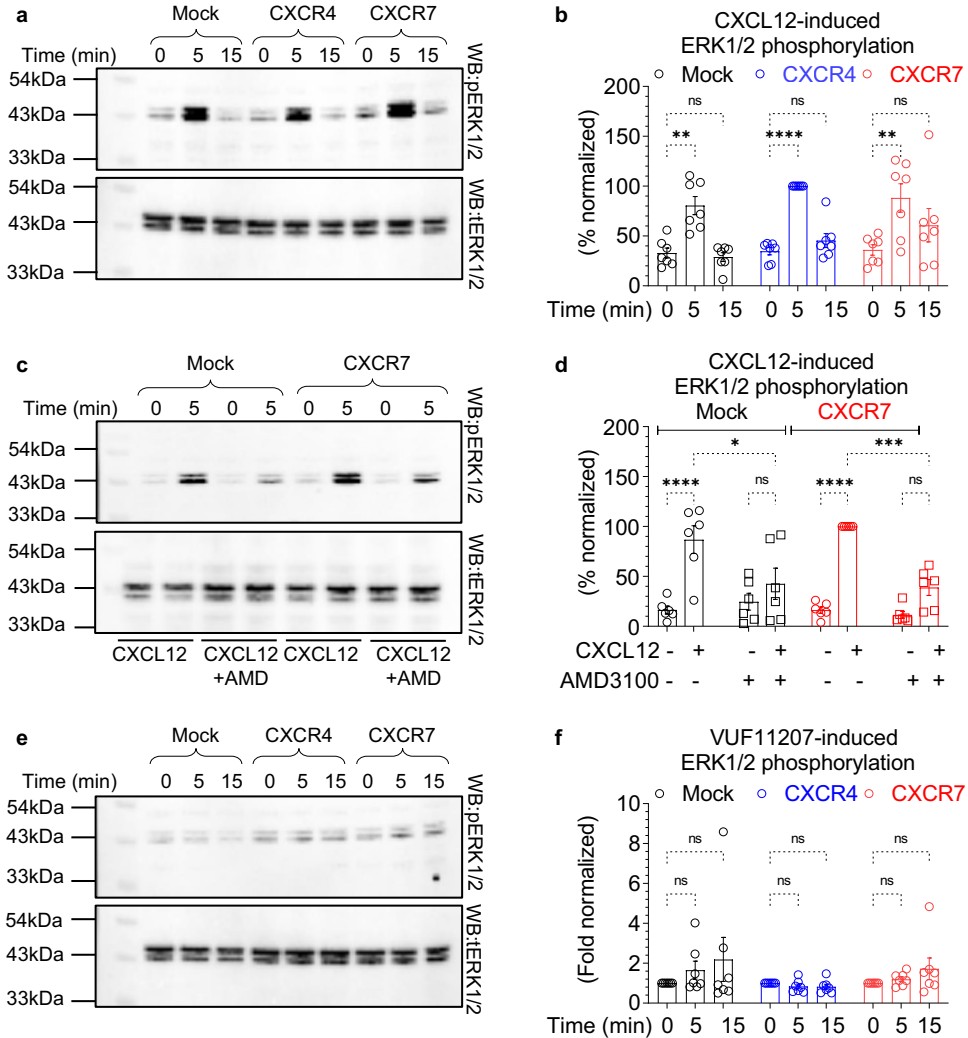

**Fig. 5 | Agonist-induced ERK1/2 phosphorylation for CXCR4 and CXCR7.**
**a**, **b** CXCL12-induced ERK1/2 phosphorylation in Mock, CXCR4 or CXCR7 transfected HEK-293 cells measured by Western blotting. Densitometry-based quantification (mean ± SEM; $n = 7$ independent experiments, normalized with the 5 min signal for CXCR4 as 100%, Two-way ANOVA, Tukey's multiple comparison test). The exact $p$-values are as follows: Mock: 0 min vs. Mock: 5 min ($p = 0.0095$), Mock: 0 min vs. Mock: 15 min ($p > 0.9999$), CXCR4: 0 min vs. CXCR4: 5 min ($p < 0.0001$), CXCR4: 0 min vs. CXCR4: 15 min ($p = 0.9940$), CXCR7: 0 min vs. CXCR7: 5 min ($p = 0.0031$), CXCR7: 0 min vs. CXCR7: 15 min ($p = 0.5552$). **c**, **d** CXCL12-induced ERK1/2 phosphorylation in CXCR7 expressing cells is blocked by pre-treatment with AMD3100 (10 μM, 30 min). Densitometry-based quantification (mean ± SEM; $n = 6$ independent experiments, normalized with CXCL12-induced signal for CXCR7 as 100%, Two-way ANOVA, Tukey's multiple comparison test). The exact $p$-values are as follows: Mock(-AMD3100): 0 min vs. Mock(-AMD3100): 5 min ($p < 0.0001$),

Mock(-AMD3100): 5 min vs. Mock(+AMD3100): 5 min ($p = 0.0192$), Mock(+-AMD3100): 0 min vs. Mock(+AMD3100): 5 min ($p = 0.8299$), CXCR7(-AMD3100): 0 min vs. CXCR7(-AMD3100): 5 min ($p < 0.0001$), CXCR7(-AMD3100): 5 min vs. CXCR7(+AMD3100): 5 min ($p = 0.0004$), CXCR7(+AMD3100): 0 min vs. CXCR7(+AMD3100): 5 min ($p = 0.3619$). **e**, **f** VUF11207-induced ERK1/2 phosphorylation in Mock, CXCR4 or CXCR7 transfected HEK-293 cells as measured by Western blotting. Densitometry-based quantification (mean ± SEM; $n = 7$ independent experiments, normalized with respect to the 0 min signal for each condition treated as 1, Two-way ANOVA, Tukey's multiple comparison test). The exact $p$-values are as follows: Mock: 0 min vs. Mock: 5 min ($p = 0.9793$), Mock: 0 min vs. Mock: 15 min ($p > 0.6170$), CXCR4: 0 min vs. CXCR4: 5 min ($p > 0.9999$), CXCR4: 0 min vs. CXCR4: 15 min ($p > 0.9999$), CXCR7: 0 min vs. CXCR7: 5 min ($p > 0.9999$), CXCR7: 0 min vs. CXCR7: 15 min ($p = 0.9633$) (*$p < 0.05$, **$p < 0.01$, ***$p < 0.001$, ****$p < 0.0001$, ns = non-significant). Source data are provided as a source data file.

## Agonist-induced ERK1/2 activation and conformations of βarr2 for CXCR4 vs. CXCR7

In order to test if βarr recruitment results in ERK1/2 MAP kinase activation, we measured agonist-induced ERK1/2 phosphorylation for CXCR4 and CXCR7 in response to CXCL12 in transfected HEK-293 cells. Although we observed robust ERK1/2 phosphorylation in CXCR4- and CXCR7-expressing cells, there was a similar response in mock-transfected cells as well (Fig. 5a, b). Moreover, ERK1/2 phosphorylation was effectively blocked by the pre-treatment of AMD3100 (CXCR4 antagonist) (Fig. 5c, d and Supplementary Fig. 6a, b). Thus, CXCL12-induced ERK1/2 phosphorylation likely arises from the endogenous CXCR4 expressed in HEK-293 cells[16]. We also note that a previous study has reported measurable ERK1/2 phosphorylation in HEK-293 cells upon CXCL12-stimulation that arises from endogenous CXCR4 and further enhanced by CXCR4 overexpression[20], which agrees with the data presented here. Furthermore, CXCL12-induced ERK1/2 phosphorylation in CXCR4-expressing cells was also completely abolished by pre-treatment with pertussis toxin (PTX) suggesting a major dependence on Gαi activation (Supplementary Fig. 6c, d). This is supported by previous studies showing that CXCL12-induced ERK1/2 phosphorylation does not require βarrs, using either siRNA-based knock-down of βarrs in HeLa cells[33], or, βarr knock-out MEFs (mouse embryonic fibroblasts)[34]. Interestingly, VUF11207 failed to elicit any measurable ERK1/2 activation at either CXCR4 or CXCR7 (Fig. 5e, f). In these experiments, the receptors were expressed at significantly higher

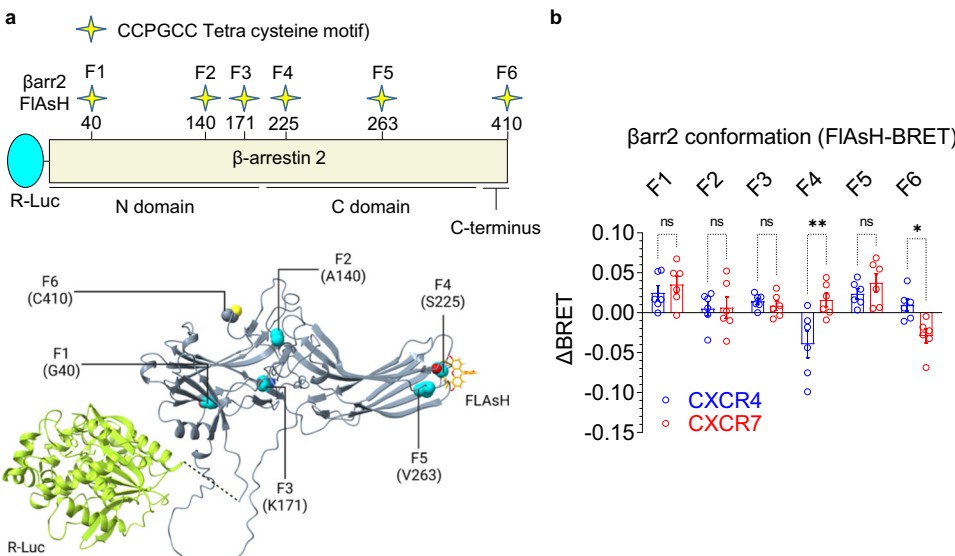

**Fig. 6 | Conformational changes in βarr2 upon interaction with CXCR4 vs. CXCR7. a** Schematic representation of intramolecular BRET-based sensors of βarr2 conformation where the N-terminus of βarr2 harbors R-Luc (Renilla luciferase) as the BRET donor, and the FlAsH motif are engineered at indicated positions in βarr2 as BRET acceptor. The structural representation in the lower panel is designed based on alpha fold generated model of βarr2. **b** CXCL12-induced BRET signal measured in HEK-293 cells expressing the indicated receptor and sensor constructs

(mean ± SEM), $n = 6$ independent experiments; the difference of the net-BRET ratio between the stimulated and unstimulated condition were plotted, Two-way ANOVA, Sidak's multiple comparison test. The exact $p$-values are as follows: F1:CXCR4 vs. F1:CXCR7 ($p = 0.9746$), F2:CXCR4 vs. F2:CXCR7 ($p > 0.9999$), F3:CXCR4 vs. F3:CXCR7 ($p = 0.9991$), F4:CXCR4 vs. F4:CXCR7 (0.0010), F5:CXCR4 vs. F5:CXCR7 ($p = 0.9052$), F6:CXCR4 vs. F6:CXCR7 ($p = 0.0420$) (*$p < 0.05$, **$p < 0.01$, ns non-significant). Source data are provided as a source data file.

level than mock-transfected cells (Supplementary Fig. 6e, f). While the lack of VUF11207-induced ERK1/2 response at CXCR4 is expected, its inability to activate ERK1/2 at CXCR7, in spite of robust βarr recruitment, is intriguing. Taken together, these data reveal that CXCR7 activation, either by CXCL12 or VUF11207, does not initiate ERK1/2 MAP kinase activation despite robust βarr recruitment. These findings also underscore the notion that βarr recruitment does not always translate to ERK1/2 activation and it may depend on specific conformation adopted by βarrs in complex with 7TMRs.

Prompted by the lack of ERK1/2 phosphorylation despite robust βarr recruitment, we decided to probe the conformation of βarr2 upon its interaction with CXCR4 and CXCR7. We used a BRET-based approach where tetra-cysteine FlAsH motifs are engineered at six distinct sites in βarr2 as BRET acceptor while the Renilla luciferase (R-Luc) is engineered at the N-terminus as BRET donor[35,36] (Fig. 6a). A change in BRET signal therefore reports conformational changes in βarr2, and a side-by-side comparison of BRET signal for these six different sensors offers readout of conformational changes in βarr2 imparted by its interaction with a receptor. Interestingly, we observed that the changes in BRET signal for at least two of these sensors were significantly different between CXCR4 and CXCR7 in response to CXCL12 (Fig. 6b). For example, in case of sensor F4 where the FlAsH label is localized between β-strand 14 and 15, there was a decrease in BRET signal for CXCR4 but an increase for CXCR7. Similarly, in case of sensor F6 where the FlAsH label is localized at position 410 in the carboxyl-terminus, there was an increase in BRET signal for CXCR4 but a decrease for CXCR7. In other sensors, the pattern of BRET change was similar between the two receptors. Taken together, these data suggest that βarr2 adopts distinct conformations upon its recruitment to CXCR4 vs. CXCR7 in response to CXCL12 although additional studies would be required to identify the precise nature of these conformational changes in more detail. Moreover, as noted earlier, CXCL11 also promotes βarr recruitment through CXCR7, and therefore, it would be interesting to compare it alongside CXCL12 and VUF11207 in terms of βarr conformation in future studies.

**Identification of the key phosphorylation-site cluster in CXCR7**

As receptor phosphorylation is a key determinant of βarr binding[27,37,38], we analyzed the carboxyl-terminus of CXCR7 and identified two distinct phosphorylation clusters, each containing three potential phosphorylation sites (Fig. 7a). These two clusters, referred to as cluster1 and 2 from here onwards, harbor PXXPXXP and PXPXXP type pattern of phosphorylation sites, respectively, where P represents a Serine or Threonine. We have previously determined the structure of βarr2 in complex with a phosphopeptide corresponding to cluster1, which revealed a partially-active βarr2 conformation in terms of the inter-domain rotation[39]. We generated two different CXCR7 constructs by mutating the phosphorylation sites in these clusters, and monitored agonist-induced βarr1/2 recruitment and endosomal localization. We observed that the mutation of cluster2 phosphorylation sites nearly abolished βarr1/2 recruitment and endosomal localization (Fig. 7b–e and Supplementary Fig. 7a–d). On the other hand, cluster1 mutation had a modest effect only on βarr1 recruitment without affecting βarr2 recruitment, and βarr1/2 endosomal localization (Fig. 7b–e). This observation potentially hints at differential contribution of specific phosphorylation sites in the recruitment of βarr isoforms, and it would be an interesting direction to investigate further in future studies. Taken together, these data establish cluster2 as the major determinant of βarr interaction and endosomal localization for CXCR7, and prompted us to investigate the contribution of individual sites in this phosphorylation cluster.

**A key phosphorylation site in CXCR7 drives βarr recruitment and endosomal localization**

We generated a series of CXCR7 mutants lacking either one or a combination of phosphorylation sites in cluster2, and tested their ability to promote βarr1/2 recruitment upon stimulation with CXCL12 and VUF11207 (Fig. 8a, b and Fig. 9a–d). We observed that the mutation of a single phosphorylation site i.e. Thr[352] resulted in a dramatic decrease in βarr1/2 recruitment, while the other two sites had relatively modest effect individually (Fig. 8a, b and Fig. 9a–d). A combination of two sites where one was Thr[352] had an additive effect in terms of

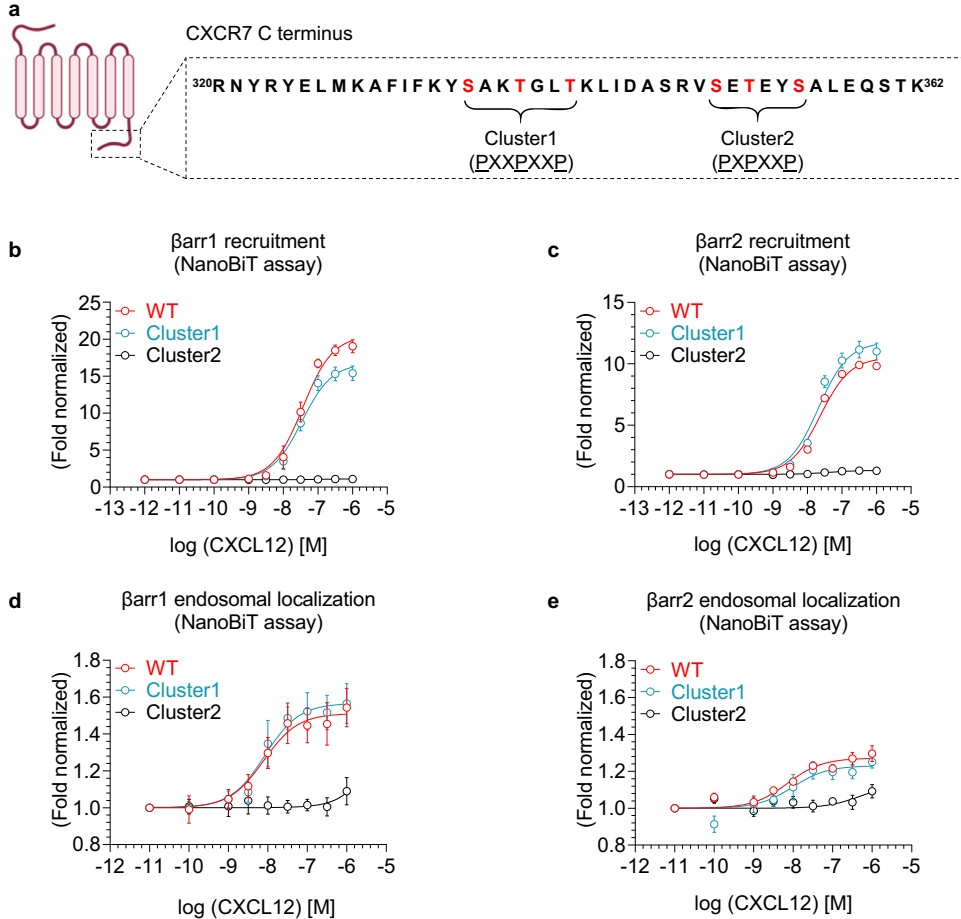

**Fig. 7 | Identification of the key phosphorylation site cluster in CXCR7. a** The carboxyl-terminus of CXCR7 harbors two potential phosphorylation site clusters indicated as cluster 1 and 2 with PXXPXXP and PXPXXP type phosphorylation codes (P is Ser/Thr; X is any other amino acid) (the schematic for the receptor is created with BioRender.com; same as in Fig. 1a/c). **b, c** Dose response curves for CXCL12-induced βarr1 and 2 recruitment to indicated CXCR7 constructs measured using the NanoBiT assay (Receptor-SmBiT+LgBiT-βarr1/2) (mean ± SEM; $n = 4$ independent experiments; normalized with luminescence signal at minimal ligand dose treated as 1). **d, e** Dose response curves of CXCL12-induced endosomal localization of βarr1/2 for the indicated CXCR7 constructs measured using the NanoBiT assay (Receptor+SmBiT-βarr1/2+LgBiT-FYVE) (mean ± SEM; $n = 4$ independent experiments; normalized with the luminescence signal at minimal ligand dose treated as 1). Source data are provided as a source data file.

attenuation in βarr binding, while the mutant lacking all three phosphorylation sites completely lost βarr recruitment. In addition to the data presented in Fig. 8a, b and Fig. 9a–d where only two saturating doses of agonists were used, we also carried out complete dose response experiments on selected mutants, which further recapitulated the same pattern of βarr1/2 recruitment (Fig. 8c, d and Fig. 10a–d). In order to test if diminished βarr recruitment also translates to reduced functional response, we measured agonist-induced endosomal localization of βarr1/2, and observed near-complete loss of endosomal localization of βarr1 and 2 for Thr[352]Ala mutation (Fig. 8e, f and Fig. 10e, f), which mirrors the recruitment pattern presented in Fig. 8c, d and Fig. 10a–d. In these experiments, the surface expression of different mutants was optimized to be comparable to the wild-type receptor (Supplementary Fig. 8a–d and Supplementary Fig. 9a–f). Taken together, these data suggest that Thr[352] is a key residue involved in directing βarr recruitment and endosomal localization for CXCR7.

## Discussion

The conceptual framework of biased agonism has focused primarily on ligands that induce distinct preferences of transducer-coupling[40–43], although biased receptor mutants have also been described for a handful of receptors[43,44]. Chemokine receptors are peculiar in this context as they display a significantly higher degree of ligand

promiscuity compared to other prototypical GPCRs, and therefore, may contain several examples of naturally-encoded ligand and receptor bias[45,46]. Our data now establish the CXCR4-CXCR7 system as an intriguing example of GPCR-ACR pair, and uncover intrinsic bias encoded at the level of transducer-coupling and functional responses. Moreover, our findings also establish VUF11207 as a highly selective agonist for CXCR7 and demonstrate that its efficacy is comparable to CXCL12, although the potency is slightly weaker.

While we comprehensively establish the lack of G-protein activation and second messenger response for CXCR7, the underlying molecular mechanism remains to be explored. For example, the atypical chemokine receptors lack the signature motifs such as the DRY and NPXXY that are present in prototypical GPCRs although reconstitution of these motifs by site-directed mutagenesis does not necessarily result in robust gain of G-protein-coupling and activation[10,11,14,47–49]. On the other hand, CXCR7 contains both, the DRY and NPXXY motif similar to prototypical GPCRs but still lacks functional G-protein-coupling. These data suggest a possible conformational mechanism wherein the agonist-induced conformation of CXCR7 is somewhat in an intermediate state with a partial opening on the intracellular side of the receptor incompatible with G-protein-coupling and/or activation[50]. It is also important to note that a recent cryo-EM structure of CXCL12-CXCR7 complex has been determined,

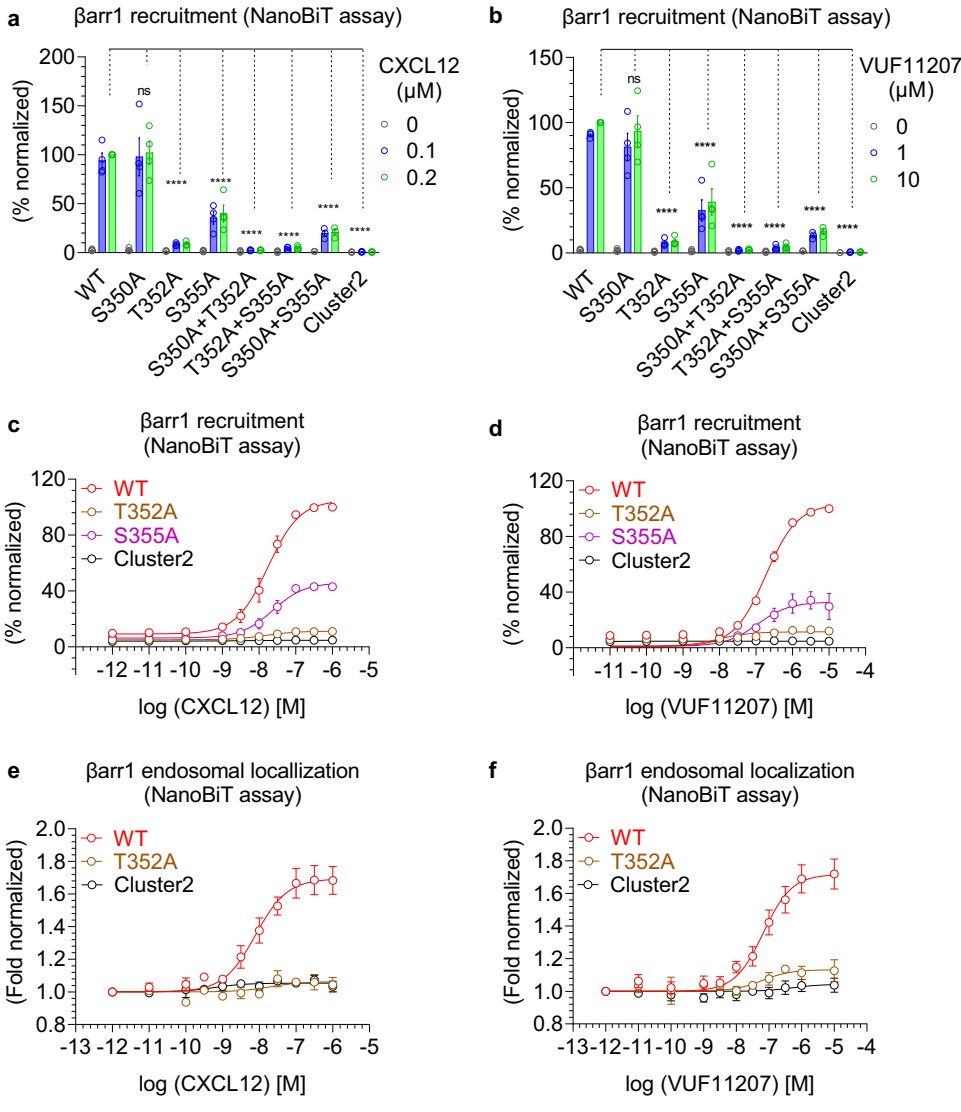

**Fig. 8 | Key phosphorylation sites in CXCR7 driving βarr1 recruitment and endosomal localization. a, b** CXCL12- and VUF11207-induced βarr1 recruitment, respectively, to the indicated phosphorylation site mutants of CXCR7 using the NanoBiT assay (mean ± SEM; $n = 3–4$ independent experiments; i.e., for CXCL12-induced βarr1 recruitment: WT, S350A, T352A, S355A, S350A + T352A, T352A + S355A, and Cluster2, $n = 4$; S350A + S355A, $n = 3$; for VUF11207-induced βarr1 recruitment: WT, S350A, T352A, S355A, S350A + T352A, T352A + S355A, and Cluster2, $n = 4$; S350A + S355A, $n = 3$; normalized with luminescence signal at maximal ligand dose for wild-type treated as 100%, Two-way ANOVA, Dunnett's multiple comparisons test. The exact $p$-values are: for CXCL12-induced βarr1 recruitment; WT vs. S350A ($p = 0.9972$), WT vs. T352A ($p < 0.0001$), WT vs. S355A ($p < 0.0001$), WT vs. S350A + T352A ($p < 0.0001$), WT vs. T352A + S355A ($p < 0.0001$), WT vs. S350A + S355A ($p < 0.0001$), WT vs. Cluster2 ($p < 0.0001$). For VUF11207-induced βarr1 recruitment; WT vs. S350A ($p = 0.4700$), WT vs. T352A ($p < 0.0001$), WT vs.

S355A ($p < 0.0001$), WT vs. S350A + T352A ($p < 0.0001$), WT vs. T352A + S355A ($p < 0.0001$), WT vs. S350A + S355A ($p < 0.0001$), WT vs. Cluster2 ($p < 0.0001$). **c, d** Dose response curves of CXCL12- and VUF11207-induced βarr1 recruitment to selected phosphorylation site mutants of CXCR7 in the NanoBiT assay (Receptor-SmBiT+LgBiT-βarr1) (mean ± SEM; $n = 3$ independent experiments; normalized with luminescence signal at maximal ligand dose for wild-type treated as 100%). **e, f** Dose response curves of CXCL12- and VUF11207-induced βarr1 endosomal localization for the selected phosphorylation site mutants of CXCR7 in the NanoBiT assay (receptor+SmBiT-βarr1 + LgBiT-FYVE) (mean ± SEM; $n = 3–7$ independent experiments; i.e., for CXCL12-induced βarr1 endosomal localization: WT, $n = 6$; T352A, $n = 3$; and Cluster2, $n = 3$; for VUF11207-induced βarr1 endosomal localization: WT, $n = 7$; T352A, $n = 3$; and Cluster2, $n = 4$; normalized with the luminescence signal at minimal ligand dose treated as 1). (****$p < 0.0001$, ns non-significant). Source data are provided as a source data file.

and the authors proposed that the positioning of ICL2 may interfere with efficient G-protein-coupling to the receptor[51]. It is also fascinating to explore whether lack of G-protein activation as demonstrated here using heterotrimer dissociation assay, and second messenger response reported in several studies, reflects an absence of physical interaction between the receptor and G-proteins or, the inability to activate G-proteins despite physical coupling. This becomes particularly important considering the reports that CXCR7 and G-proteins may exist in close proximity in cellular context although agonist-induced increase in their proximity and/or interaction was not apparent[51]. Finally, we cannot rule out the contribution of cellular

context such as presence of specific lipid environment, accessory proteins and post-translational modifications that may also contribute to the lack of G-protein activation by CXCR7 measured in HEK-293 cells.

It is also noteworthy that despite a robust recruitment of βarrs, CXCR7 fails to elicit any measurable ERK1/2 phosphorylation in response to either CXCL12 or VUF11207. Previous studies using a combination of direct binding assays and biophysical methods have correlated βarr conformation and activation with the interaction of c-Raf1, MEK-1 and ERK2 binding, and ERK2 phosphorylation in-vitro[52,53]. Thus, it is plausible that the lack of ERK1/2 activation downstream of

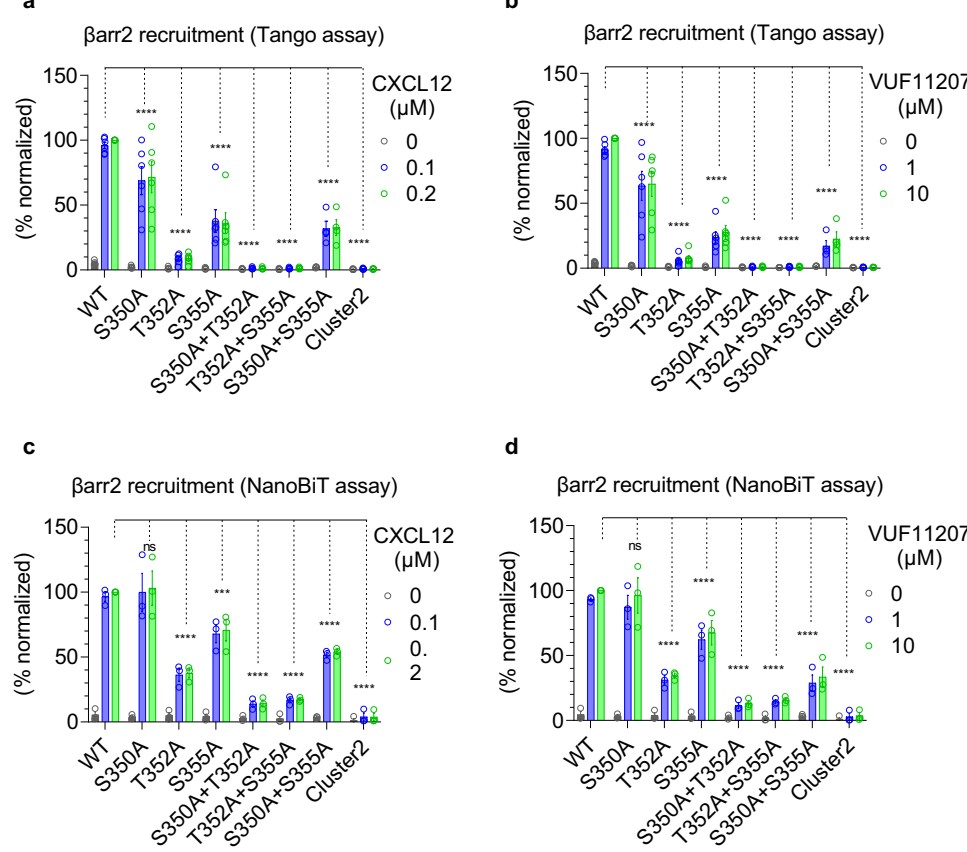

**Fig. 9 | Contribution of different phosphorylation sites in CXCR7-mediated βarr1/2 recruitment. a**, **b** CXCL12- and VUF11207-induced βarr2 recruitment, respectively, to the indicated phosphorylation site mutants of CXCR7 using the Tango assay (mean ± SEM); $n = 4$–6 independent experiments; i.e., for CXCL12: WT, S350A, T352A, S355A, S350A + T352A, T352A + S355A, and Cluster2, $n = 6$; S350A + S355A, $n = 4$; for VUF11207-induced βarr2 recruitment: WT, S350A, T352A, S355A, S350A + T352A, T352A + S355A, and Cluster2, $n = 6$; S350A + S355A, $n = 4$; normalized with luminescence signal at maximal ligand dose for wild-type treated as 100%, Two-way ANOVA, Dunnett's multiple comparisons test. The exact $p$-values are: for CXCL12-induced βarr2 recruitment; WT vs. S350A ($p < 0.0001$), WT vs. T352A ($p < 0.0001$), WT vs. S355A ($p < 0.0001$), WT vs. S350A + T352A ($p < 0.0001$), WT vs. T352A + S355A ($p < 0.0001$), WT vs. S350A + S355A ($p < 0.0001$), WT vs. Cluster2 ($p < 0.0001$). For VUF11207; WT vs. S350A ($p < 0.0001$), WT vs. T352A ($p < 0.0001$), WT vs. S355A ($p < 0.0001$), WT vs. S350A + T352A ($p < 0.0001$), WT vs. T352A +

S355A ($p < 0.0001$), WT vs. S350A + S355A ($p < 0.0001$), WT vs. Cluster2 ($p < 0.0001$). **c**, **d** CXCL12- and VUF11207-induced βarr2 recruitment, respectively, to the indicated phosphorylation site mutants of CXCR7 using the NanoBiT assay (mean ± SEM); $n = 3$ independent experiments; normalized with luminescence signal at maximal ligand dose for wild-type treated as 100%, Two-way ANOVA, Dunnett's multiple comparisons test. The exact $p$-values are: for CXCL12; WT vs. S350A ($p = 0.9994$), WT vs. T352A ($p < 0.0001$), WT vs. S355A ($p = 0.0001$), WT vs. S350A + T352A ($p < 0.0001$), WT vs. T352A + S355A ($p < 0.0001$), WT vs. S350A + S355A ($p < 0.0001$), WT vs. Cluster2 ($p < 0.0001$). For VUF11207; WT vs. S350A ($p = 0.8957$), WT vs. T352A ($p < 0.0001$), WT vs. S355A ($p < 0.0001$), WT vs. S350A + T352A ($p < 0.0001$), WT vs. T352A + S355A ($p < 0.0001$), WT vs. S350A + S355A ($p < 0.0001$), WT vs. Cluster2 ($p < 0.0001$) (***$p < 0.001$, **** $p < 0.0001$, ns non-significant). Source data are provided as a source data file.

CXCR7 despite robust βarr recruitment reflects a βarr conformation that is not conducive to scaffolding and/or activation of ERK1/2 MAP kinase. In fact, our data with intramolecular BRET sensor of βarr2 hints at distinct conformations in βarr2 induced by CXCR4 vs. CXCR7. A change in BRET for an intramolecular sensor typically reflects a change in the distance and/or orientation of the donor and acceptor moieties, and therefore, directionally opposite changes in BRET signal for CXCR4 vs. CXCR7 for the same sensor suggest a difference in distance and/or orientation of the FlAsH sites relative to the N-terminus (i.e. R-Luc fusion site). However, direct structural visualization of βarrs in complex with CXCR4 and CXCR7 at high resolution is required to better understand the precise conformational differences, and how they are linked to resulting functional outcomes for these two receptors. Taken together, these data support the notion that βarr-binding does not always translate into ERK1/2 activation at 7TMRs, and receptor-bound conformations of βarrs fine-tune the resulting signaling outcomes. The identification of Thr[352] as a key site driving CXCR7-βarr recruitment and endosomal localization of βarrs presents an interesting paradigm where the spatial positioning of a single site is

also critical in determining βarr binding and functional outcomes, in addition to the phosphorylation bar-code as suggested previously[54–57].

In summary, our study provides molecular insights into the intrinsic-bias encoded in the CXCR4-CXCR7 system, and present an important advance to better understand the intricate details of 7TMR-βarr interaction and signaling with potential therapeutic implications.

## Methods

### General reagents, plasmids, and cell culture

Most of the general reagents were purchased from Sigma Aldrich unless mentioned otherwise. Dulbecco's Modified Eagle's Medium (DMEM), Fetal-Bovine Serum (FBS), Dulbecco's Phosphate buffer saline (PBS), Trypsin-EDTA, Hank's balanced salt solution (HBSS), and penicillin-streptomycin solution were purchased from Thermo Fisher Scientific. HEK-293 cells (ATCC) and HTLA cells were maintained in DMEM (Gibco, Cat. no. 12800-017) supplemented with 10% FBS (Gibco, Cat. no. 10270-106) and 100 U ml⁻¹ penicillin (Gibco, Cat.

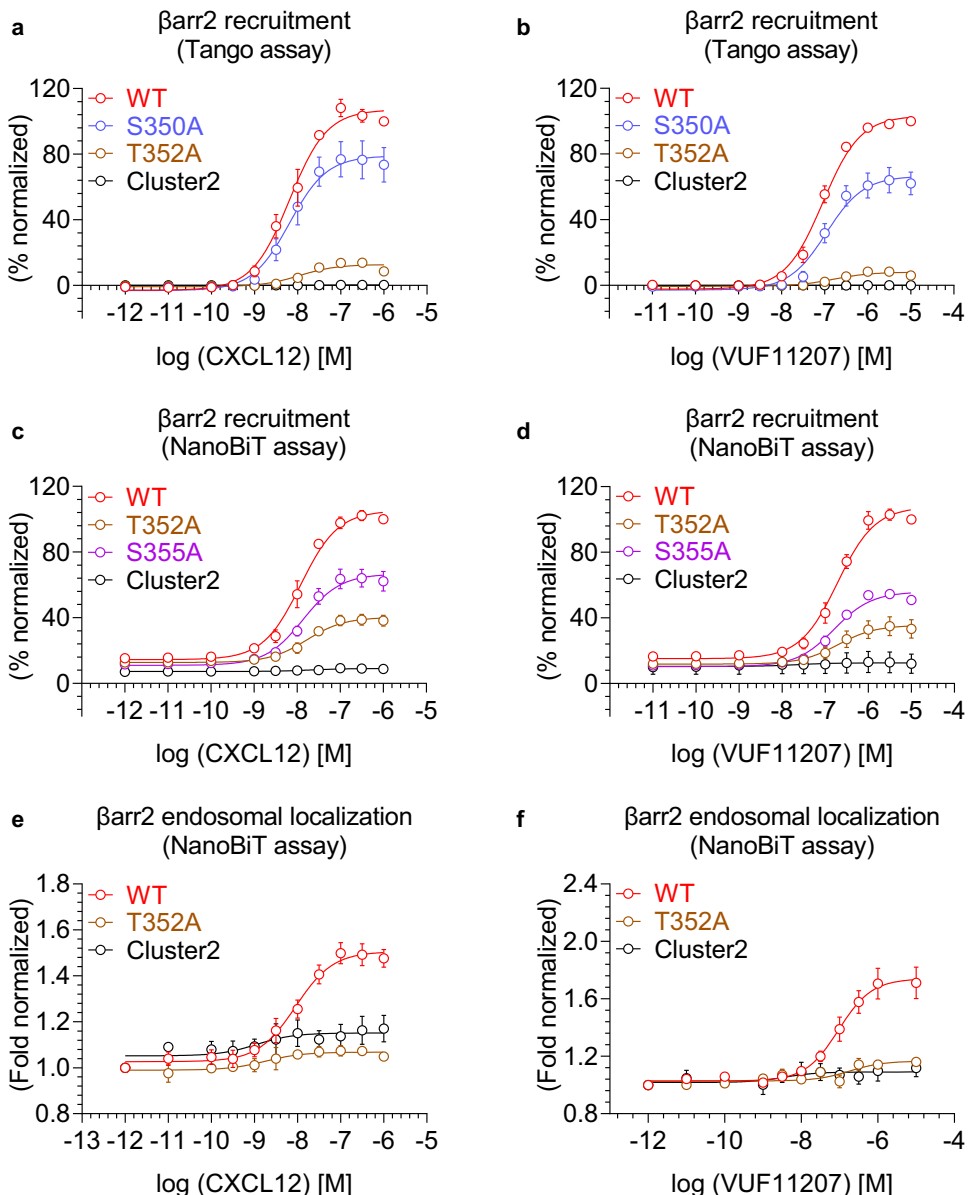

**Fig. 10 | Effect of Thr352Ala mutation in CXCR7 on βarr2 recruitment and endosomal localization. a–d** Dose-response curves of CXCL12- and VUF11207-induced βarr2 recruitment to selected phosphorylation site mutants of CXCR7 in the Tango and NanoBiT assays (mean ± SEM; $n = 6$ independent experiments for **a**, **b** and $n = 3$ independent experiments for **c**, **d**; normalized with luminescence signal for WT at maximal ligand dose treated as 100%). **e**, **f** Dose-response curves of CXCL12- and VUF11207-induced βarr2 endosomal localization for the selected

phosphorylation site mutants of CXCR7 in the NanoBiT assay (Receptor+SmBiT-βarr2 + LgBiT-FYVE) (mean ± SEM; $n = 3$–7 independent experiments; i.e., for CXCL12-induced βarr2 endosomal localization: WT, $n = 6$; T352A, $n = 3$; and Cluster2, $n = 3$; for VUF11207-induced βarr2 endosomal localization: WT, $n = 7$; T352A, $n = 3$; and Cluster2, $n = 4$; normalized with the luminescence signal at minimal ligand dose treated as 1). Source data are provided as a source data file.

no. 15140122) and 100 μg ml$^{-1}$ streptomycin (Gibco, Cat. no. 15140-122) at 37 °C in 5% $CO_2$. The cDNA coding region for CXCR1-7 were cloned in pcDNA3.1 with an N-terminal FLAG tag and an HA signal sequence. PRESTO-Tango assay constructs were acquired from Addgene (Cat. no. 1000000068) while the Tango assay constructs were generated in our laboratory[58]. CXCR7 phosphorylation site mutants were generated by site-directed mutagenesis using Q5 Site-Directed Mutagenesis Kit (NEB, Cat. no. E0554S). For the NanoBiT assay, receptor constructs with carboxyl-terminus SmBiT were generated. For nanoBiT based βarr recruitment assay, large fragment (LgBiT) of the nanoluciferase enzyme was fused at the N-terminus of βarr. To study endosomal localization, βarr was tagged with small fragment (SmBiT) of nanoluciferase and large fragment (LgBiT) was fused to FYVE domain of human endofin with a flexible linker. For

G-protein dissociation assay, helical domain of Gα subunit was tagged with large fragment (LgBiT) and small fragment (SmBiT) was fused at the N-terminus of Gγ. All constructs were verified by DNA sequencing (Macrogen). The oligonucleotide primers used in this study are listed in Supplementary Table 1. Recombinant CXCL12 was purchased from PeproTech (Cat. no. 300-28 A), and VUF11207 was from Sigma (Cat. no. SML0669). In addition, we also synthesized VUF11207 following a previously published protocol[18] and used it in some experiments. The antibodies used in this study were HRP-conjugated anti-FLAG M2 (1:2,000; Sigma-Aldrich, Cat. no. A8592), anti-pERK1/2 (1:5,000; Cell Signaling Technology, Cat. no. 9101), anti-tERK1/2 (1:5,000; Cell Signaling Technology, Cat. no. 9102), and HRP-coupled anti-rabbit IgG secondary antibody (1:10,000 dilution; Genscript, Cat. No. A00098).

## Synthesis of VUF11207

The schematic of VUF11207 synthesis is shown in the Supplementary Fig. 10. A detailed protocol of VUF11207 synthesis has been published earlier[18]. Briefly, Aldol condensation of 2-fluorobenzaldehyde with propionaldehyde under basic condition gave (E)-3-(2-Fluorophenyl)-2-methylacrylaldehyde (1) in 72% yield. In the presence of acetic acid, aldehyde 1 reacted with 2-(1-methylpyrrolidin-2-yl) ethanamine in methanol, and the resulting imine was reduced with NaBH(OAc)3 to generate (E)-3-(2-fluorophenyl)-2-methyl-N-(2-(1-methylpyrrolidin-2-yl)ethyl) prop-2-en-1-amine (2) in 50% yield. Under the standard amide coupling conditions (EDCI/HOBt/DIEA), amine 2 was coupled with 3,4,5-trimethoxybenzoic acid to afford VUF11207 with 67% yield. The structures of target compound and the intermediates were confirmed by their spectral properties.

## NanoBiT-based G-protein dissociation assay

Ligand-induced G-protein activation was measured using NanoBiT-based G-protein dissociation assay[19]. Briefly, a NanoBiT-G-protein consisting of Gα subunit tagged with LgBiT and SmBiT-tagged Gγ2 subunit along with the untagged Gβ1 subunit were co-expressed with the indicated receptor constructs and ligand-induced change in luminescence signal was measured. Typically, HEK-293 (Thermo Fisher Scientific) cells were transfected with a plasmid mixture consisting of 100 ng Gα-LgBiT, 500 ng Gβ1, 500 ng SmBiT-Gγ2 (C68S) with either 50 ng of CXCR4 plasmid or 3.5 μg of CXCR7. The receptor constructs used in this assay contain N-terminal HA signal sequence, FLAG tag and a flexible linker sequence. To enhance NanoBiT-G-protein expression for Gs, Gq and G12/13, 100 ng of RIC8B plasmid (isoform 2; for Gs) or RIC8A (isoform 2; for Gq, G12, and G13) were also co-transfected. 24 h post-transfection, cells were harvested with EDTA-containing PBS, centrifuged, and suspended in 2 ml of HBSS (Gibco, Cat. no. 14065-056) containing 0.01% bovine serum albumin (BSA fatty acid–free grade, SERVA) and 5 mM 4-(2-hydroxyethyl)-1-piperazineethanesulfonic acid (HEPES), pH 7.4 (assay buffer). Afterwards, cells were dispensed in a white 96-well plate (80 μl well$^{-1}$), incubated with 20 μl of 50 μM coelenterazine (Carbosynth, Cat. no. EC175526), and 2 h later, baseline luminescence was measured (SpectraMax L, Molecular Devices). Subsequently, 20 μl of 6X agonist, serially diluted in the assay buffer, were manually added and the plate was immediately read for the second measurement in a kinetic mode. Luminescence counts recorded from 3-5 min post-agonist addition were averaged, corrected with the baseline signals, normalized with respect to vehicle control plotted using the GraphPad Prism 9.5.0 software.

## GloSensor-based cAMP assay

In order to assess agonist-induced coupling of Gαs and Gαi, we used GloSensor-based cAMP assay[59]. Briefly, HEK-293 cells were co-transfected with FLAG-tagged receptor constructs and luciferase-based 22 F cAMP biosensor (Promega, Cat. no. E2301) using poly-ethylenimine (PEI) linear (Polysciences, Cat. no. 19850) at a ratio of 1:3 (DNA:PEI linear) as transfection reagent. After 14-16 h of transfection, cells were detached from the plates, resuspended in assay buffer (1XHBSS, 20 mM HEPES, pH 7.4) containing D-luciferin (0.5 mg ml$^{-1}$, GoldBio, Cat. no. LUCNA-1G) and seeded into 96 well white plates (Corning) at a density of $2 \times 10^5$ cells well$^{-1}$ in a volume of 100 μl. After an incubation of 1.5 h at 37 °C and 30 min at room-temperature, baseline luminescence readings were recorded. For Gαs-coupling assay, ligands prepared in the assay buffer were added at indicated final concentration after baseline readings while for Gαi-coupling assay, 5 μM forskolin (Sigma, Cat. no. F6886) was added to the cells and luminescence readings were recorded till they stabilized (5-10 cycles) followed by ligand addition. The change in luminescence signal was recorded using a microplate reader (Victor X4; Perkin Elmer) for 60 min, and data were normalized as indicated in the respective figure legends and plotted using GraphPad Prism 9.5.0 software. For the experiments presented in Fig. 1e, 0.25 μg of CXCR4 and 5 μg of CXCR7 were used along with 2 μg of 22 F cAMP biosensor. For the Gαs-coupling assay (Fig. 1f), 0.5 μg of CXCR4, 0.5 μg of V$_2$R and 4 μg of CXCR7 plasmids were used along with 3 μg of 22 F cAMP biosensor. For the experiments presented in Fig. 3c, 3.5 μg of CXCR1, CXCR3, and CXCR4 plasmids were used while 5 μg of plasmids were used for CXCR2, CXCR5, CXCR6, and CXCR7 along with 2 μg of 22 F cAMP biosensor.

## Calcium flux assay

HEK-293 cells were transiently transfected with pGP-CMV-GCaMP6s (Ca$^{2+}$ Sensor plasmid, Addgene, Cat. no. 40753; 4 μg), 5HT$_{2c}$ receptor (as positive control, cDNA.org, Cat. no. HTR02CTN00; 4 μg) or CXCR4/CXCR7 receptor (4 μg) using PEI max in a ratio of 1:4 (DNA:PEI max) and plated at a density of $5 \times 10^4$ cells well$^{-1}$ in black optical bottom plate in complete DMEM media (10% FBS). After 14-16 h of transfection, media from the plate was aspirated and 100 μl of Ca$^{2+}$/Mg$^{2+}$ free HBSS buffer (pH 7.2) was added, cells were further incubated at 37 °C for 10 min in the Flex Station 3 (Molecular Devices) before the assay was initiated. Ligand induced change in relative fluorescence unit (RFU) was measured at excitation 485 nm and emission 525 nm (cut off 515 nm) with the settings of 6 reads well$^{-1}$. Basal fluorescence of each well was recorded for 15 s, and then 20 μl of 6X concentration of each agonist (Serotonin and VUF11207) as indicated was added using robotic pipetting of FlexStation system and RFU was recorded at 2 s interval for a total of 135 s. The changes in RFU (ΔRFU) for each treatment group was calculated by subtracting the average basal response (RFU before ligand addition) from RFU of each well at each time points after ligand addition. ΔRFU for each ligand was plotted and analyzed using GraphPad Prism 9.5.0 software.

## Receptor surface expression

In order to measure the surface expression of the receptors in various assays, we used a previously described whole cell-based surface ELISA assay[60]. Transfected cells from the corresponding assays were seeded into a 24-well plate pre-coated with 0.01% poly-D-Lysine at a density of $2 \times 10^5$ cells well$^{-1}$ and incubated at 37 °C for 24 h. Afterwards, cells were washed once with ice-cold 1XTBS, fixed with 4% PFA (w/v in 1XTBS) on ice for 20 min, washed again three times with 1XTBS, and blocked at room temperature for 1.5 h with 1% BSA prepared in 1XTBS. Subsequently, the cells were incubated with anti-FLAG M2-HRP antibody (Sigma, Cat. no. A8592) (1:2,000 for 1.5 h at room temperature) followed by three washes in 1% BSA and incubation with TMB-ELISA substrate (Thermo Fisher Scientific, Cat. no. 34028) until the light blue color appeared. The signal was quenched by transferring 100 μl of the colored solution to another 96 well plate containing 100 μl of 1 M H$_2$SO$_4$, and the absorbance was measured at 450 nm. For normalization, TMB substrate was removed, cells were washed twice with 1XTBS, and incubated with 0.2% (w/v) Janus Green (Sigma, Cat. no. 201677) for 15 min at room temperature. The excess stain was removed by washing the cells with water followed by addition of 800 μl of 0.5 N HCl in each well, and 200 μl of this solution was transferred to a 96 well plate for measuring the absorbance at 595 nm. The signal intensity was normalized by calculating the ratio of A450/A595 values and plotted using the GraphPad Prism 9.5.0 software. The surface expression data in every experiment were normalized with respect to mock-transfection (i.e. pcDNA) treated as 1.

For the NanoBiT-based G-protein dissociation assay, surface expression of the receptors was measured using flow-cytometry based method. Briefly, a small amount of HEK-293 cells from the corresponding assays were harvested with 0.5 mM EDTA-containing PBS and transferred to a 96 well V-bottom plate. Cells were fluorescently labeled using anti-FLAG monoclonal antibody (Clone 1E6, FujiFilm Wako Pure Chemicals; 10 μg ml$^{-1}$ diluted in 2% goat serum + 2 mM EDTA-containing PBS) followed by incubation with Alexa Fluor

488-conjugated goat anti-mouse IgG secondary antibody (Thermo Fisher Scientific; 10 µg ml$^{-1}$). Subsequently, the cells were washed with PBS, resuspended in 2 mM EDTA-containing PBS, filtered through a 40 µm filter and the fluorescent intensity of single cells was quantified using a flow cytometer. Fluorescent signal from Alexa Fluor 488 was recorded and analyzed using the FlowJo software. Mean fluorescence intensity from about 20,000 cells per sample were used for analysis.

## Tango assay for βarr recruitment

In order to assess the βarr2 recruitment to indicated receptors, Tango assay was used[61]. Briefly, HTLA cells were transfected with indicated receptor constructs and 24 h post-transfection, cells were trypsinized, resuspended in complete DMEM, and seeded into 96 well white plates at a density of $1 \times 10^5$ cells well$^{-1}$. After another 24 h, cells were stimulated with the indicated dose of ligands and incubated at 37 °C for additional 7-8 h. Afterwards, the culture media was changed with assay buffer (1XHBSS, 20 mM HEPES, pH 7.4 and 0.5 mg ml$^{-1}$ D-luciferin). Luminescence readings were measured in a microplate reader (Victor X4; Perkin Elmer), normalized as mentioned in the corresponding figure legends, and plotted using GraphPad Prism 9.5.0 software. For the data presented in Figs. 2a, c, and 3a and Supplementary Fig. 2d, PRESTO-Tango constructs were used. PRESTO-Tango constructs harbors V$_2$R tail at the end of the native receptor followed by a TEV cleavage site and tTA transcription factor. For the data presented in Figs. 2b, d and 3b, Tango assay constructs were generated by engineering a TEV protease cleavage site and tTA transcription factor at the end of the receptor coding sequence in pcDNA3.1 vector backbone.

## NanoBiT assay for βarr recruitment

Agonist-induced βarr1/2 recruitment for CXCR4 and CXCR7 was also measured using NanoBiT-based assay[62]. Briefly, HEK-293 cells were transfected with CXCR4 (1 µg) and CXCR7 (7 µg) harboring carboxyl-terminus fusion of SmBiT and βarr1/2 constructs (2 µg) with N-terminal fusion of LgBiT. The cells were stimulated with varying doses of respective ligands followed by measurement of luminescence signal using a multimode plate reader for 10-15 cycles and average data from 5$^{th}$ to 10$^{th}$ cycle are used for analysis and presentation. In order to evaluate the contribution of different GRKs in βarr recruitment to CXCR7, we used previously described GRK knock-out cell lines[31].

## Microfluidic chemotaxis assay

We quantified chemotaxis using a microfluidic device that tracks movement of single cells toward a gradient[63] and we used MDA-MB-231 human breast cancer cells (purchased from the ATCC, Manassus, VA, USA) stably transduced with CXCR7 fused to GFP[64]. Briefly, we introduced MDA-MB-231 cells stably transduced with CXCR7 fused to GFP into the device at a concentration of $1 \times 10^5$ cells ml$^{-1}$ in complete DMEM medium with 10% serum and 1% GlutaMAX. After allowing cells to adhere for 10 min, we replaced medium in the seeding port with serum-free DMEM. We added a chemoattractant, 100 ng ml$^{-1}$ CXCL12 (R&D Systems) and/or 100 nM VUF11207 (Cayman Chemical) in serum-free DMEM with 0.1% Probumin (Millipore), to the opposite side of the device. We quantified chemotaxis of single cells after 16 h in the device.

## ERK1/2 MAP kinase phosphorylation assay

Agonist-induced ERK1/2 MAP kinase phosphorylation was measured using the Western blot assay[65,66]. Briefly, HEK-293 cells were transfected with CXCR4 (0.25 µg), CXCR7 (4 µg) or empty vector (pcDNA3.1; 7 µg), and 24 h post-transfection, they were seeded into a 6 well plate at a density of $1 \times 10^6$ cells well$^{-1}$. Subsequently, the cells were serum starved for 12 h followed by agonist-stimulation (100 nM CXCL12 and 10 µM VUF11207) as indicated in the corresponding figure legends. Afterwards, the cells were harvested, lysed in 2XSDS loading buffer, heated at 95 °C for 15 min followed by centrifugation at 21000 x g for 15 min. 10 µl of lysate was then separated by SDS-PAGE and ERK1/2

bands were detected by Western blotting using corresponding antibodies (rabbit phospho-ERK1/2 antibody, 1:5,000 dilution; rabbit total ERK1/2 antibody, 1:5,000 dilution; anti-rabbit HRP-coupled secondary antibody, Genscript, Cat. No. A00098, 1:10,000 dilution). ECL solution from Promega (Cat. no. W1015) was used as a substrate for the HRP, and the signals were developed using ChemiDoc (BioRad). The signals were quantified using densitometry in BioRad Image Lab software, normalized as indicated in the figure legend, and data were plotted using GraphPad Prism 9.5.0 software. For the experiments presented in Fig. 5c, d and Supplementary Fig. 6a, b, cells were pre-treated with 10 µM of AMD3100 for 30 min prior to CXCL12 stimulation, and for the experiments presented in Supplementary Fig. 6c, d, cells were pre-treated with Pertussis toxin (100 ng µl$^{-1}$) for 12 h during serum starvation step.

## BRET assay for βarr2 conformational change

Intramolecular FlAsH-based BRET sensors were used to monitor the conformational changes in βarr2[35,36]. Briefly, HEK-293SL cells were seeded at a density of $1.5 \times 10^5$ cells well$^{-1}$ in 6 well plates and transfected with the indicated receptor constructs along with the βarr2-FlAsH sensors using calcium phosphate. 24 h post-transfection, cells were detached and seeded into poly-ornithine-coated 96 well white plates at a density of $2.5 \times 10^4$ cells well$^{-1}$. After another 24 h, cells were washed and incubated with Tyrode's buffer (140 mM NaCl, 2.7 mM KCl, 1 mM CaCl$_2$, 12 mM NaHCO$_3$, 5.6 mM D-glucose, 0.5 mM MgCl$_2$, 0.37 mM NaH$_2$PO$_4$, 25 mM HEPES, pH 7.4) for 1 h at room temperature. Subsequently, FlAsH reagent solution was prepared by mixing 1.75 µl of FlAsH-EDT2 stock reagent with 3.5 µl of 25 mM EDT solution in DMSO and left for 10 min at room temperature. 100 µl of Tyrode's buffer was added to this mixture followed by an additional incubation for 5 min at room temperature and then the volume was adjusted to 5 ml with Tyrode's buffer. Cells were incubated with 60 µl of the labeling solution for 1 h at 37 °C followed by washing with BAL wash buffer and Tyrode's buffer. Finally, 90 µl of Tyrode's buffer was added to each well and the plate was incubated at 37 °C for 1 h before ligand stimulation. Coelenterazine H was added at a final concentration of 2 µM, cells were stimulated with 100 nM CXCL12 and six consecutive BRET measurements were taken using a Victor X; PerkinElmer plate reader with a filter set (center wavelength/band width) of 460/25 (donor) and 535/25 (acceptor). BRET ratios (intensity of light emitted by the acceptor/intensity of light emitted by the donor) were calculated and net-BRET ratio was determined after subtracting the background BRET ratio i.e. the difference between the FlAsH-EDT2-labeled BRET ratio and the unlabeled condition. The difference of the net-BRET ratio for ligand-stimulated condition vs. vehicle-treatment was plotted using GraphPad Prism 9.5.0 software.

## βarr recruitment for the phosphorylation site mutants of CXCR7

The phosphorylation site mutants of CXCR7 as indicated in Fig. 8a, b and Fig. 9a–d were generated using Q5 site-directed mutagenesis kit followed by βarr recruitment in Tango and NanoBiT assays. Surface expression of the indicated mutants was first optimized to be at comparable levels followed by the Tango and NanoBiT assays. For the Tango assay, HTLA cells were transfected with 7 µg of the wild-type and mutant receptor constructs except the CXCR7$^{T352A+S355A}$ for which, 5 µg DNA was transfected. Cells were treated with indicated concentration of agonists for 8 h at 37 °C followed by the addition of luciferin and luminescence was recorded. Ligand-induced luminescence signal was normalized with respect to the minimal ligand dose concentration taken as 1 and plotted in GraphPad Prism 9.5.0.

## NanoBiT assay for βarr endosomal localization

Agonist-induced βarr1/2 endosomal localization was monitored using NanoBiT assay. HEK-293 cells were transiently transfected with receptor, N-terminal SmBiT-tagged βarr1/2 constructs and N-terminal

LgBiT-tagged FYVE constructs. The amount of DNA for receptor, βarr1/2 and FYVE was kept as 7 µg, 2 µg and 5 µg, respectively.

## Data quantification and statistical analysis

All the experiments described here were carried out at least three times and data (mean ± SEM) are plotted and analyzed using GraphPad Prism software (9.5.0). The data were normalized with respect to proper experimental controls and appropriate statistical analyses were performed as indicated in the corresponding figure legends.

## Reporting summary

Further information on research design is available in the Nature Portfolio Reporting Summary linked to this article.

## Data availability

All the relevant data are included in the manuscript and the Supplementary Information files. Source data are provided with this paper. Any additional information can be obtained from the corresponding authors upon request. Source data are provided with this paper.

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

## Acknowledgements

Research in A.K.S.'s laboratory is supported by the Senior Fellowship of the DBT Wellcome Trust India Alliance (IA/S/20/1/504916) awarded to A.K.S., Science and Engineering Research Board (EMR/2017/003804, SPR/2020/000408, and IPA/2020/000405), Council of Scientific and Industrial Research [37(1730)/19/EMR-II], Indian Council of Medical research (F.NO.52/15/2020/BIO/BMS), Young Scientist Award from Lady Tata Memorial Trust, and IIT Kanpur. H.D.A. was supported by a BioCare grant from DBT (BT/PR31791/BIC/101/1228/2019). A.I. was funded by the LEAP JP20gm0010004, and the BINDS JP20am0101095 from the Japan Agency for Medical Research and Development (AMED); Japan Society for the Promotion of Science (JSPS) KAKENHI grants 21H04791, 21H05113, JPJSBP120213501 and JPJSBP120218801; FOREST Program JPMJFR215T and JST Moonshot Research and Development Program JPMJMS2023 from Japan Science and Technology Agency (JST); The Uehara Memorial Foundation; and Daiichi Sankyo Foundation of Life Science. This work was also supported by a grant from the Canadian Institutes of Health Research (CIHR) (MOP-74603) to S.A.L., and Y.C. is supported by a doctoral training scholarship from the Fonds de recherche santé Québec. The research work in XC's laboratory is supported by a grant from the National Science Foundation of China (21272029). G.D.L. is supported by United States NIH grants R01CA238023, U24CA237683, R01CA238042, U01CA210152, R33CA225549, and R37CA222563. K.E.L. is supported by NIH grant R50CA221807. We thank Manish K. Yadav, Jagannath Maharana and Ramanuj Banerjee for helping with manuscript preparation and revision, Annu Dalal and Nashra Zaidi for helping with surface expression assays, and Manisankar Ganguly for preparing the structural schematic in Fig. 6a.

## Author contributions

P.S. performed the cAMP assays, βarr recruitment and endosomal localization assays, ERK1/2 phosphorylation assay, site-directed mutagenesis with help from D.S., S.P., H.D.-A.; C.M.C.C. and K.K. performed the G-protein dissociation and βarr recruitment in GRK knock-out cells under the supervision of A.I.; X.R. synthesized VUF11207 under the supervision of X.C., Y.-C.C. carried out the BRET experiment under the supervision of S.A.L.; P.K. performed the Calcium assay with the guidance from P.N.Y.; K.E.L. generated cell lines used for migration experiments performed by Y.-C.C. under supervision of G.D.L.; A.K.S. supervised and managed the overall project; all authors contributed to data analysis, interpretation and manuscript writing.

## Competing interests

G.D.L. and K.E.L. receive research funding from InterAx AG administered through the University of Michigan. All other authors declare no competing interests.
