## [Peer Review File · Nature Communications]

Molecular insights into intrinsic transducer-coupling bias in the CXCR4-CXCR7 systemEditorial Note: This manuscript has been previously reviewed at another journal that is not operating a transparent peer review scheme. This document only contains reviewer comments and rebuttal letters for versions considered at *Nature Communications*.

REVIEWERS' COMMENTS

Reviewer #1 (Remarks to the Author):

The authors have responded to my concerns, but unfortunately the removal of the crystal structure further decreases the distinction between this work and the existing extensive prior work on the CXCR4/CXCR7 system. As currently presented the key new information (CXCR7 phosphosite and tool compound for CXCR7) are buried within a lot of other data that is largely confirmatory.

Reviewer #2 (Remarks to the Author):

The current manuscript analyzed different cellular signaling of CXCR7 and CXCR4 with two different ligands using well-designed cell system and studied the differences between CXCR7- and CXCR4-mediated conformational changes of b-arrestins. The authors also defined the key phosphorylation sites in CXCR7 for b-arrestin recruitment. The work was thoroughly done, but I have a few comments to improve the manuscript.

1. Throughout the figures, the fonts are not consistent within the same figure. In the reviewer's experience, this could be due to different font mechanism of a specific analysis program (for example, GraphPad Prism) between different countries. Please correct these.
2. In figure 2B, the plot for CXCR4 seems missing.
3. The authors identified the key phosphorylation residues in CXCR7. Would these phosphorylation residues make CXCR7 to strongly interact with b-arrestins than CXCR4? Are the phosphorylation residues in CXCR4 defined? Or, would the core-engagement make CXCR7 to strongly interact with b-arrestins than CXCR4?

Reviewer #3 (Remarks to the Author):

In general, the authors have nicely responded to my comments on the previous version of this manuscript. My specific comments concerning the revised manuscript or rebuttal follow below.

I agree with the authors that this study advances our understanding of CXCR7 at multiple levels, and establishes VUF11207 as a synthetic chemical probe at CXCR7. I'm hesitant to call it an agonist without further validation, but nevertheless, VUF11207 holds the potential to uncover new insights into CXCR7 biology. This is exciting, but without these new insights the advance here is somewhat incremental.

VUF11207 promoting migration of MDA-MB-231-CXCR7 cells is interesting. However, I'm still concerned regarding the phenomenological nature of these data. It is well known that GPCR over-expression can impact coupling/signaling, therefore this type of experimental result, while potentially exciting, requires additional vetting. For example, does CXCR7, like CXCR4, switch coupling in the MDA-MB-231 cells? Is the migration sensitive to pertussis toxin? Have you examined lines with different levels of stable expression of CXCR7? What about the CXCR7 phospho-mutants that don't recruit Barr?

Also, while I believe that the specific contribution of GRKs in Barr recruitment, conformational signatures in Barr2 and identification of key phosphorylation sites in CXCR7 driving Barr recruitment are important findings and will be of interest to the field, the advance here is more or less descriptive rather than mechanistic in nature. While these results are new for CXCR7, in my opinion, they are more or less predictable compared to similar data with other GPCRs that this group has studied. I appreciate the author's response, and I don't want to minimize their contributions here, but I'm having difficulty appreciating the conceptual advance.

I appreciate the author's attempts to normalize receptor expression by increasing the DNA amount of CXCR7 in transfections (Fig. R6; Fig. SD-E & S3E-F). It is interesting that CXCR7 seems to be under-expressed at the cell surface when compared with CXCR4 or V2R. This probably speaks to differences in how these GPCRs are regulated in terms of stability on the cell surface and/or trafficking to the plasma membrane. In future experiments, I would suggest transfecting less of the GPCR that is not limiting to achieve equal surface expression. For CXCR4 especially, lower is better and might be important, since CXCR4 is well known to self-associate or self-aggregate in experimental settings that require its heterologous expression. I can envision how a PRESTO-tango, BRET, Nano-bit Barr recruitment assay can be influenced by this.

Normalizing data to mock within each experiment is fine, but there has to be an error associated with this, unless the mock readings are naturally identical, which I doubt. I suggest the authors consult a statistical expert for help.

Reference: NCOMMS-23-12711-T

Response to reviewers:

Reviewer #1

The authors have responded to my concerns, but unfortunately the removal of the crystal structure further decreases the distinction between this work and the existing extensive prior work on the CXCR4/CXCR7 system. As currently presented the key new information (CXCR7 phosphosite and tool compound for CXCR7) are buried within a lot of other data that is largely confirmatory.

We sincerely thank the reviewer for her/his positive comments on our manuscript. We have included all the key data in the main figures and further underscored the key findings in the corresponding sections.

Reviewer #2:

The current manuscript analyzed different cellular signaling of CXCR7 and CXCR4 with two different ligands using well-designed cell system and studied the differences between CXCR7- and CXCR4-mediated conformational changes of b-arrestins. The authors also defined the key phosphorylation sites in CXCR7 for b-arrestin recruitment. The work was thoroughly done, but I have a few comments to improve the manuscript.

We sincerely thank the reviewer for her/his positive comments on our manuscript.

1. Throughout the figures, the fonts are not consistent within the same figure. In the reviewer's experience, this could be due to different font mechanism of a specific analysis program (for example, GraphPad Prism) between different countries. Please correct these.

We have noticed that the fonts were getting changed during PPPT to PDF conversion due to some technical issue. We have now uploaded the final figures directly as power point files.

2. In figure 2B, the plot for CXCR4 seems missing.

We have double checked this figure and the plot for CXCR4 is present. Perhaps, it was not visible in the PDF version of the figures due to technical reasons.

3. The authors identified the key phosphorylation residues in CXCR7. Would these phosphorylation residues make CXCR7 to strongly interact with b-arrestins than CXCR4? Are the phosphorylation residues in CXCR4 defined? Or, would the core-engagement make CXCR7 to strongly interact with b-arrestins than CXCR4?

This is an interesting point. While there have been some studies on CXCR4 with respect to phosphorylation sites and β arr recruitment, more elaborate studies are warranted in future to pinpoint the precise contribution of specific sites. In addition, future studies are also required to probe the formation of core vs. tail-engaged complexes between CXCR4/CXCR7 and β arrs, and we have discussed these points in the revised manuscript (line 153-158, page 7).

Reviewer #3:

In general, the authors have nicely responded to my comments on the previous version of this manuscript. My specific comments concerning the revised manuscript or rebuttal follow below.

We thank the reviewer for her/his positive comments on our manuscript and appreciating our extensive efforts to revise the manuscript.

I agree with the authors that this study advances our understanding of CXCR7 at multiple levels, and establishes VUF11207 as a synthetic chemical probe at CXCR7. I'm hesitant to call it an agonist without further validation, but nevertheless, VUF11207 holds the potential to uncover new insights into CXCR7 biology. This is exciting, but without these new insights the advance here is somewhat incremental.

We thank the reviewer for her/his positive comments on our work, and agreeing with us about the significant advances that our study provides towards understanding CXCR7 biology at multiple levels. Our reasoning to call VUF11207 as an agonist of CXCR7 is based on its ability to induce β arr recruitment and promote endosomal localization. We have mentioned this explicitly in the revised manuscript (line 172-174, page 8).

VUF11207 promoting migration of MDA-MB-231-CXCR7 cells is interesting. However, I'm still concerned regarding the phenomenological nature of these data. It is well known that GPCR over-expression can impact coupling/signaling, therefore this type of experimental result, while potentially exciting, requires additional vetting. For example, does CXCR7, like CXCR4, switch coupling in the MDA-MB-231 cells? Is the migration sensitive to pertussis toxin? Have you examined lines with different levels of stable expression of CXCR7? What about the CXCR7 phospho-mutants that don't recruit Barr?

We thank the reviewer for making these interesting suggestions, which we may take up in future studies as we believe that they are beyond the scope of the current study. We have added a sentence in the revised manuscript (line 186-188, page 8) to acknowledge this point.

Also, while I believe that the specific contribution of GRKs in Barr recruitment, conformational signatures in Barr2 and identification of key phosphorylation sites in CXCR7 driving Barr recruitment are important findings and will be of interest to the field, the advance here is more or less descriptive rather than mechanistic in nature. While these results are new for CXCR7, in my opinion, they are more or less predictable compared to similar data with other GPCRs that this group has studied. I appreciate the author's response, and I don't want to minimize their contributions here, but I'm having difficulty appreciating the conceptual advance.

We thank the reviewer for appreciating the novelty and impact of our work. As the reviewer has mentioned, our study provides several important insights into distinct transducer-coupling patterns between CXCR4 and CXCR7, and they should catalyze additional studies in future to better understand the mechanistic aspects of intrinsic-bias encoded in this receptor system.

I appreciate the author's attempts to normalize receptor expression by increasing the DNA amount of CXCR7 in transfections (Fig. R6; Fig. SD-E & S3E-F). It is interesting that CXCR7 seems to be under-expressed at the cell surface when compared with CXCR4 or V2R. This probably speaks to differences in how these GPCRs are regulated in terms of stability on the cell surface and/or trafficking to the plasma membrane. In future experiments, I would suggest transfecting less of the GPCR that is not limiting to achieve equal surface expression. For CXCR4 especially, lower is better and might be important, since CXCR4 is well known to self-associate or self-aggregate in experimental settings that require its heterologous expression. I can envision how a PRESTO-tango, BRET, Nano-bit Barr recruitment assay can be influenced by this.

We thank the reviewer for appreciating our efforts to normalize the receptor expression, and we will consider reviewer's suggestion in future studies.

Normalizing data to mock within each experiment is fine, but there has to be an error associated with this, unless the mock readings are naturally identical, which I doubt. I suggest the authors consult a statistical expert for help.

We are normalizing with respect to mock condition (treated as 1 or 100%) in each experiment, and therefore, there is no error bar associated to this condition across experimental replicates. We have clarified this point in the corresponding figure legends.